# Development of the Long-term Harmonized multi-satellite SIF (LHSIF) dataset at 0.05° resolution (1995–2024)

Chu Zou[1,2,3], Shanshan Du[1,2], Xinjie Liu[1,2], Liangyun Liu[1,2,3]

[1] Key Laboratory of Digital Earth Science, Aerospace Information Research Institute, Chinese Academy of Sciences, Beijing, 100094, China

[2] University of Chinese Academy of Sciences, Beijing 100049, China

[3] International Research Center of Big Data for Sustainable Development Goals, Beijing 100094, China

*Correspondence to: Liangyun Liu (liuly@radi.ac.cn)*

**Abstract.** Solar-induced chlorophyll fluorescence (SIF) is a crucial proxy of photosynthetic processes in vegetation. In recent decades, advancements in remote sensing technology have facilitated long-term global SIF monitoring, significantly enhancing our understanding of vegetation dynamics on a global scale. Despite this progress, current SIF datasets face major challenges, including temporal inconsistencies among various satellite-derived products and a lack of long-term, high-resolution observations. In this study, we developed a "Long-term Harmonized SIF" (LHSIF) dataset spanning 1995 to 2024 with a fine spatial resolution of 0.05° by coordinating SIF satellite observations from GOME, SCIAMACHY, GOME-2, and OCO-2. Light use efficiency (LUE)-based spatial downscaling models were employed for each SIF product to generate fine-resolution global SIF maps. The long-term dataset was constructed using temporally corrected GOME-2A SIF (TCSIF) as a benchmark and was combined with a cumulative distribution function (CDF) normalization method for far-red SIF harmonization across satellite sensors from GOME, SCIAMACHY, and OCO-2. The resulting harmonized dataset shows a 49% reduction in inter-sensor differences compared to the uncorrected data and exhibits a stable interannual increase of $0.31 \pm 0.07\%$ yr$^{-1}$. This result strongly aligns with the growth rate of gross primary production (GPP, $0.47 \pm 0.03\%$ yr$^{-1}$) and is consistent with ground-based SIF observations (R > 0.60). Therefore, the long-term harmonized SIF dataset with a fine 0.05° resolution is valuable for estimating global photosynthesis over extended periods. The LHSIF dataset is available at https://doi.org/10.5281/zenodo.16394372 (Zou et al., 2025).

**Keywords:** Solar-induced chlorophyll fluorescence; Multi-satellite harmonization; Spatial downscaling

## 1  Introduction

Solar-induced chlorophyll fluorescence (SIF) is an optical signal naturally released by plants, closely linked to their photosynthetic dynamics (Zhang et al., 2016; Zhang and Peñuelas, 2023; Zhu et al., 2024; Rascher et al., 2015; Porcar-Castell et al., 2014; Damm et al., 2015; Mohammed et al., 2019). SIF has garnered significant attentions due to its potential as a novel proxy for gross primary productivity (GPP) (Ryu et al., 2019), bridging the gap in our understanding of global photosynthetic processes (Beer et al., 2010; Anav et al., 2015; Chen et al., 2024).

Following the publication of the initial global SIF map from the Greenhouse Gases Observing Satellite (GOSAT), interest in the SIF-GPP association greatly increased (Frankenberg et al., 2011; Guanter et al., 2012; Joiner et al., 2011). Subsequent satellite-based analyses have consistently revealed strong spatial and temporal correlations between SIF and GPP, showcasing remarkable alignment between SIF and GPP in terms of spatial distribution and seasonal variability (Anav et al., 2015; Li et al., 2018; Verma et al., 2017; Yang et al., 2015; Guanter et al., 2014; Zheng et al., 2024). However, these results are mostly based on coarse-resolution SIF datasets such as the Global Ozone Monitoring Experiment (GOME)-2, leading to potential spatial mismatch issues. Additionally, the SIF-GPP link varies by vegetation type, emphasizing the critical need for SIF datasets with higher spatial resolution and spatiotemporal consistency to better support ecosystem monitoring and interpretation.

Long-term global SIF observations are important for analyzing the vegetation functions and changes under different climatic conditions. Multiple high-spectral-resolution satellite missions have provided publicly available global SIF products since 1995. The earliest records originated from the GOME sensor on European Remote sensing Satellite (ERS) in 1995, followed by the SCanning Imaging Absorption spectroMeter for Atmospheric CartograpHY (SCIAMACHY) onboard Environmental Satellite (EnviSat) in 2003. However, these sensors had relatively short operational lifespans, ceasing operations in 2003 and 2012, respectively. The GOME-2 sensor onboard the MetOp-A satellite, launched in January 2007, operated until November 2021; this sensor provided the longest SIF time series to date (Joiner et al., 2013). The Orbiting Carbon Observatory(OCO)-2 satellite, launched in 2014, features exceptionally high spatial resolution and has been validated through synchronized airborne campaigns (Sun et al., 2017) to ensure the reliability of resulting SIF products. Recent studies highlight the potential of the TROPOMI sensor onboard Sentinel-5P (Koren et al., 2018; Wen et al., 2020), but its SIF products are currently constrained to a relatively short time series (May 2018 to April 2021).

Despite the availability of multiple satellite SIF products, most have a temporal coverage shorter than 10 years, and large discrepancies have been observed between different SIF products (Parazoo et al., 2019). These temporal inconsistencies may stem from differences in retrieval algorithms, absolute radiometric calibration errors, instrumental artifacts, directional effects, and variations in satellite overpass times and footprint sizes (Zhang et al., 2018c; Bacour et al., 2019). To address these challenges, Wen et al. (2020) proposed a harmonization framework that used the cumulative distribution function (CDF) to integrate SIF datasets from SCIAMACHY and GOME-2 during their overlapping period, resulting in a continuous record from 2002 to 2018.

While Wen's framework laid the foundation for cross-sensor harmonization, it did not explicitly address instrument

degradation—a key factor that compromises the long-term consistency of single-sensor records. Such degradation, as observed in GOME-2, poses a significant challenge for long-term consistency and introduces uncertainties in trend analyses (Parazoo et al., 2019). For instance, Yang et al. (2018) reported diverging trends between EVI and SIF, attributing the latter's decline to reduced photosynthetic activity. However, Zhang et al. (2018a) argued that this conclusion was impacted by the deterioration of the GOME-2A instrument. Further research by Koren et al. (2018) showed that the decline in SIF persisted even after correcting for sensor degradation. The SIFTERv2 product (van Schaik et al., 2020) employed in Koren's study was simply corrected using linear models; the reliability of SIFTERv2 decreased significantly after 2016, limiting its application for long-term trend analysis.

To mitigate this limitation, Wang et al. (2022) attempted to create a temporally corrected long-term SIF product (LT_SIFc*) by correcting the degradation trends in gridded GOME, SCIAMACHY, and GOME-2 SIF products. However, the method lacks a physically based correction of the actual sensor radiance degradation and instead applies adjustments on the SIF product, which may not accurately reflect the true instrumental change. This is further complicated by the nonlinear characteristics inherent in the SIF retrieval methodology and subsequent processing procedures (e.g., zero-bias correction and quality filtering), which prevent a direct and linear propagation of sensor degradation into the final SIF retrievals. Recently, the temporally corrected GOME-2A SIF dataset (TCSIF) included a calibration of the radiance measurements of GOME-2A using a pseudo-invariant method (Zou et al., 2024). This correction effectively eliminates the influence of sensor degradation over time, providing a robust benchmark for generating long-term harmonized SIF products.

So far, the cross-sensor consistency of existing long-term SIF records remains to be further evaluated. In this study, we employed the TCSIF dataset as a physically calibrated benchmark to constrain the long-term consistency of GOME, SCIAMACHY, and OCO-2 SIF observations. By harmonizing these multi-sensor datasets against a radiometrically corrected reference, we generated a continuous and temporally consistent SIF product spanning from 1995 to 2024, which is the longest multi-satellite harmonized SIF dataset to date. Additionally, we performed light use efficiency (LUE)-based spatial downscaling on the coarse spatial resolution dataset derived from the satellite SIF products. This downscaling reduced the spatial difference between satellite-derived SIF and ground-based measurements of SIF and GPP, thereby facilitating our understanding of vegetation photosynthesis at the global scale.

## 2 Method and materials

### 2.1 Satellite-based SIF datasets

#### 2.1.1 GOME SIF

GOME, which was launched in 1995 on the ERS-2 satellite of the European Space Agency (ESA), was initially developed to measure the column densities of ozone and nitrogen dioxide (Hahne et al., 1993). GOME's channel 4 operates within a spectral range of 590 – 790 nm, achieving a spectral resolution of about 0.5 nm for far-red SIF retrieval. Although GOME is characterized by a relatively low spatial resolution of 320 km × 40 km, it provides the earliest available record of SIF data. The GOME SIF product utilized in this research is the daily averaged SIF signal at 740 nm, which is retrieved by data-driven algorithms (Joiner et al., 2019). The dataset spans the period from July 1995 to June 2003.

### 2.1.2   SCIAMACHY SIF

The SCIAMACHY instrument was in operation from 2002 to 2012 onboard ESA's Envisat satellite, overlapping with the timeframe of GOME. The instrument enhanced GOME's capabilities by offering a finer spatial resolution of 30 km × 60 km. The comparable spectral ranges and spectral resolutions of SCIAMACHY and GOME allowed for the use of analogous techniques for SIF retrievals. The SCIAMACHY SIF products we used were retrieved using the same data-driven algorithms and fitting window (734–758 nm) as those used for GOME. Daily SCIAMACHY SIF datasets at 740 nm from January 2003 to April 2012 were employed (Joiner et al., 2021).

### 2.1.3   GOME-2A SIF

As a successor to GOME, GOME-2 is part of EUMETSAT's MetOp satellite series, with three satellites (MetOp-A, B, and C) launched between 2007 and 2018. GOME-2 improved upon its predecessor by providing enhanced spatial resolution (40 km × 40 km or 80 km × 40 km, contingent upon the specific platform utilized). The GOME-2A SIF datasets were obtained from the MetOp-A satellite, launched in 2007 and operating until 2021.

Research has shown apparent differences between GOME-2A SIF products using different retrieval methods (Parazoo et al., 2019). For instance, SIF retrieval using a fitting window of 720–758 nm and a backward elimination algorithm (Köhler et al., 2015) yields values up to twice as large as the retrievals using a 734–758 nm window (Joiner et al., 2013). The GOME-2A SIF dataset used in this study was retrieved using the same data-driven algorithm and fitting window as Joiner et al. (2013), ensuring consistency with GOME and SCIAMACHY SIF. Furthermore, the GOME-2 SIF we use has undergone correction for sensor degradation and was found to avoid spurious trends caused by instrument deterioration (Zou et al., 2024). Therefore, this temporal-corrected GOME-2A SIF dataset with degradation correction is used as a benchmark to harmonize the data from the other three sensors.

### 2.1.4   OCO-2 SIF

OCO-2 was a satellite mission launched by the National Aeronautics and Space Administration in 2014. Unlike earlier missions, OCO-2 focuses on small target areas, attaining a considerably greater spatial resolution of approximately 1.3 km × 2.3 km. The spectral range of OCO-2 extends from 757 to 775 nm, facilitating the initial SIF retrievals at 757 nm and 771 nm. Drawing upon the empirical correlation of SIF across various wavelengths, the product offers daily global SIF at 740 nm (OCO-2/OCO-3 Science Team, 2020). The SIF datasets from OCO-2 and GOME-2 have eight overlapping years (2014 to 2021). As a result, a thorough comparison and validation of the consistency can be conducted between the two datasets.

The product specifications and sensor information are listed in Table 1. This study resampled the orbital SIF data from different satellites into global gridded datasets of varying sizes according to the footprint and the global coverage of the satellites. Satellite-derived SIF measurements from GOME, SCIAMACHY, GOME-2, and OCO-2 were aggregated into monthly maps with grid sizes of 1°×1°, 1°×1°, 0.5°×0.5°, and 1°×1°, respectively.

**Table 1 Information on multiple satellite SIF datasets used to construct long-term SIF products.**

| Satellite/Sensor | Temporal range | Footprint Size (km²) | Overpass time | Swath width (km) | Wavelength (nm) | Grid size | Reference |
| --- | --- | --- | --- | --- | --- | --- | --- |
| ERS-2/ GOME | 1995.07–2003.06 | 40×320 | 10: 30 | 960 | 740 | 1°×1° | (Joiner et al., 2019) |
| Envisat/ SCIAMACHY | 2003.01–2012.04 | 30×240 /30×60 | 10: 00 | 960 /240 | 740 | 1°×1° | (Joiner et al., 2021) |
| MetOp-A/ GOME-2 | 2007.01–2021.11 | 40×80 /40×40 | 9: 30 | 1920 /960 | 740 | 0.5°×0.5° | (Zou et al., 2024) |
| OCO-2 | 2014.09–2024.12 | 1.3×2.2 | 13: 30 | 10.3 | 757/771 | 1°×1° | (OCO-2/OCO-3 Science Team, 2020) |

## 2.2 Spatial downscaling

An LUE-based model was used for downscaling the gridded SIF datasets with coarse spatial resolutions. Assuming that SIF can be represented using the LUE model in a manner that is comparable to GPP (Berry et al., 2012; Guanter et al., 2014; Damm et al., 2015), then:

$$\text{SIF} = \text{PAR} \times \text{fPAR} \times SIF_{yield} \tag{1}$$

where $SIF_{yield}$ is the fluorescence quantum yield, which is influenced by hydric and thermic stresses. fPAR represents the fraction of photosynthetically active radiation (PAR) that is absorbed by vegetation, which exhibits a positive correlation with vegetation indices. Assuming that PAR is uniformly distributed over small areas and can be considered constant, then equation (1) can be further expressed as (Duveiller and Cescatti, 2016; Duveiller et al., 2020) :

$$\text{SIF} \approx b_0 \times \text{f(NIRv)} \times \text{f(VPD)} \times \text{f}(AT) \tag{2}$$

where NIRv is the near-infrared reflectance of vegetation, VPD represents vapor pressure deficit (accounts for the effect of hydric stresses), and $AT$ is the air temperature at 2 m (accounts for the impact of thermic stresses). A quadratic function, sigmoid function, and Gaussian function with unknown coefficients were used to express f(NIRv), f(VPD), and f($AT$), respectively, as follows:

$$SIF \approx b_1 NIRv^{b_2} \times \left[ \frac{1}{(1 + \exp(b_3(b_4 - VPD)))} \right] \times \left[ \exp\left( -0.5 \left( \frac{AT + b_5}{b_6} \right)^2 \right) \right]. \tag{3}$$

The unknown coefficients $b_1$ to $b_6$ in Eq. (3) can be determined by a nonlinear iterative approach. Here, we implemented this approach using the "L-BFGS-B" algorithm (Byrd et al., 1995).

NIRv, VPD, and AT were the three driving variables of the spatial downscaling model. NIRv datasets characterized by a spatial resolution of 0.05° were partially derived from the Advanced Very High Resolution Radiometer (AVHRR) (Jeong et al., 2024) for 1995–2021, while the NIRv for 2022–2024 were calculated using MODIS MCD43C4 nadir reflectance (Schaaf and Wang., 2021). AT and VPD data for 1995–2024 were obtained from the TerraClimate product with an original spatial resolution of 1/24° (Abatzoglou et al., 2018). The driving variables were aggregated to coarse spatial resolutions (0.5°×0.5° for GOME-2 and 1°×1° for other satellites) for training the LUE model described in Eq. (3) and to estimate the coefficients ($b_1$ to $b_6$). The 50 closest neighbors of the center pixel were utilized to train the LUE model in an 11×11 sliding window. Subsequently, SIF datasets, characterized by a spatial resolution of 0.05°, were produced by inputting the 0.05° driving variables into the trained model. Since the coefficients ($b_1$ to $b_6$) were computed for each coarse-resolution pixel, gridded artifacts may appear in the final product. To further ensure smooth spatial transitions, for each high-resolution pixel within a coarse-resolution pixel, a 3×3 block of coarse-resolution pixels was selected, and nine sets of six coefficients ($b_1$ to $b_6$) were computed. Meanwhile, a fine-resolution (0.05°) weighting grid was established within the 3×3 low-resolution pixels using a two-dimensional Gaussian function with a standard deviation of 15 km. The final downscaled result for each high-resolution pixel was obtained as the weighted average of the nine sets of model-predicted values, following the approach of Duveiller and Cescatti (2016).

## 2.3 CDF matching method

The cross-sensor SIF normalization was implemented using a stratified CDF matching approach to account for environmental variability. Specifically, the stratification was done based on a combination of Köppen climate zones (Beck et al., 2023) and the MODIS land cover types product (MCD12C1; Friedl and Sulla-Menashe, 2022). The degradation-corrected GOME-2A dataset was used as the normalization reference for all other satellite-derived SIF datasets, based on their overlapping periods. The normalization of GOME data was based on the SCIAMACHY dataset, which had been previously normalized with GOME-2 data.

For each sensor pair, the cumulative distribution functions of both the reference and target datasets were calculated across their overlapping temporal coverage. A linear interpolation was used to match the quantiles of the target dataset with those of the reference. Separate CDF transfer functions were derived for each calendar month to account for phenological variations. For SCIAMACHY and GOME, due to the limited temporal overlap, the entire period (from January 2003 to June 2003) was used to construct the CDF function. The complete workflow is shown in Fig. 1.

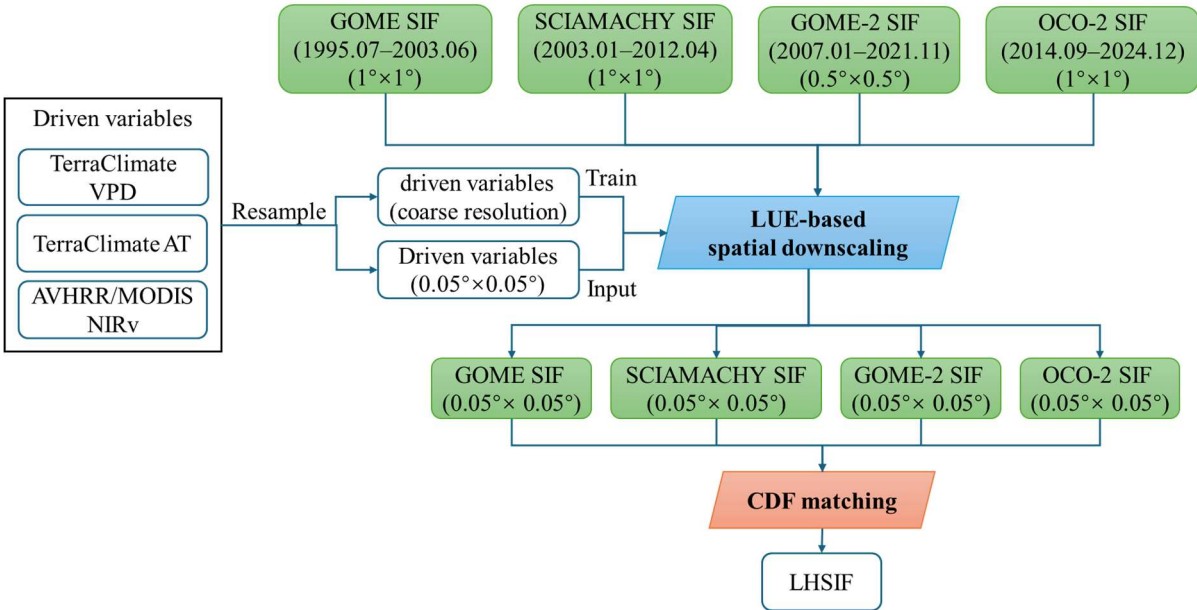

**Figure 1. The workflow for spatial downscaling and temporal alignment for generating the LHSIF products from 1995 to 2024.**

Further, the discrepancy between the two SIF time series can be quantified by the mean squared difference (MSD):

$$\text{MSD} = \frac{1}{n}\sum_{i=1}^{n}\left(D_{1_i} - D_{2_i}\right)^2 \tag{5}$$

where $D_1$ and $D_2$ are the SIF time series of the two SIF datasets to be compared. i represents the i-th month of the chosen period. Furthermore, Eq. (1) can be broken down into three terms (Bacour et al., 2019):

$$\text{MSD} = \left(\overline{D_1} - \overline{D_2}\right)^2 + \left(\sigma_{D_1} - \sigma_{D_2}\right)^2 + 2\,\sigma_{D_1}\,\sigma_{D_2}(1 - r) \tag{6}$$

where $\overline{D_1}$ and $\overline{D_2}$ are the expected values of the time series, while $\sigma_{D_1}$ and $\sigma_{D_2}$ signify the respective standard deviations. Additionally, r is the Pearson correlation coefficient that quantifies the relationship between the datasets. The first and second terms in the formula represent the square of the mean deviation (denoted as $bias^2$) and difference in standard deviation (denoted as $variance^2$) between the corrected datasets and the target datasets. The final term quantifies the inconsistency of the linear correlation between the two datasets (denoted as phase).

## 2.4 Temporal Trend Analysis Metrics

To assess long-term trends in vegetation dynamics, we employed the Mann-Kendall (MK) test, a non-parametric method

suitable for detecting monotonic trends in time series data, using the Python package pyMannKendall (Hussain and Mahmud, 2019). Trend estimation uncertainty was quantified by the standard deviation and 95% confidence intervals of the estimated temporal trend.

**2.5  Datasets for validation and comparison analysis**

**2.5.1 Boreal Ecosystem Productivity Simulator GPP**

The Boreal Ecosystem Productivity Simulator (BEPS) is an ecological process model that integrates vegetation parameters with meteorological data to simulate ecosystem productivity. We used the GPP dataset generated by the BEPS model for 1995–2019 (Weimin Ju and Zhou, 2021). The original spatial resolution of the dataset is $0.072727° × 0.072727°$, providing fine-scale insights into productivity dynamics. This high-resolution dataset allows for detailed spatiotemporal analysis and facilitates comparisons with downscaled SIF datasets in this study to help our understanding of ecosystem carbon dynamics.

**2.5.2 Long-term satellite SIF products**

The LT_SIFc* dataset provides long-term SIF retrievals corrected for temporal inconsistencies between GOME, SCIAMACHY, and GOME-2 SIF datasets (Wang et al., 2022). A CDF method was employed for the harmonization of different SIF datasets, and the LUE-based model was used for spatial downscaling. The LT_SIFc* dataset spans 1995 to 2018 at a spatial resolution of $0.05° × 0.05°$.

The SIF_005 dataset is a SIF product spanning 2003 to 2017, with a spatial resolution of $0.05° × 0.05°$ (Wen et al., 2020). This product integrates data from SCIAMACHY and GOME-2 SIF datasets, and it is downscaled using a machine learning-based method. The v2.2 (trend_corrected) version was utilized in this study; the original SIF dataset used for this version has been preliminarily corrected for temporal degradation.

The LCSIF dataset provides global SIF estimates from 1982 to 2022 at $0.05° × 0.05°$ resolution, derived from bias-corrected AVHRR and MODIS reflectance data (Fang et al., 2023). A neural network (NN) model was trained to predict OCO-2 SIF using two surface reflectance bands (red and near-infrared), after inter-sensor radiometric calibration between AVHRR and MODIS during their overlapping period.

**2.5.3 AVHRR vegetation indices**

Global NDVI and NIRv datasets from 1995 to 2021, derived from the AVHRR sensors, were utilized in this study. These datasets  were developed by Jeong et al. (2024) based on the AVHRR Long-Term Data Record version 5 (LTDR V5) surface reflectance product. To address temporal inconsistency in long-term AVHRR records, a three-step correction was applied,

including cross-sensor calibration, orbital drift correction, and machine learning-based harmonization with MODIS vegetation indices. This post-processing significantly improved the temporal consistency of NDVI and NIRv from 1982 to 2021, as verified using detrended anomalies and trends at calibration sites. The final product enables more robust analyses of long-term vegetation dynamics and reduces spurious trends due to sensor artifacts.

### 2.5.4 Ground-based observations

Ground-based SIF and GPP observations were integrated into this study to validate and enhance the interpretation of satellite-derived datasets. Specifically, FLUXNET GPP observations were employed, which are based on in-situ measurements from a global network of flux towers distributed across diverse ecosystems (Pastorello et al., 2020). FLUXNET sites with more than five years of data were grouped into climate zones and vegetation functional types (see Fig. S1 for site distribution and types). The field "GPP_DT_VUT_REF" was used. To ensure the quality of the GPP data used for validation, only GPP records with the quality flag greater than 0.7 (Verma et al., 2015) were retained in this study.

In addition, tower-based SIF observations from the ChinaSpec network, including sites such as DM, GC, HL, XTS, and AR (Zhang et al., 2021), were used to validate the accuracy and spatiotemporal consistency of the long-term SIF dataset generated in this study. The locations and cover types of the ChinaSpec sites used are listed in Table S1. To ensure consistent comparisons, the tower-based SIF at 760.6 nm was converted to 740 nm using an empirical correction factor of 1.48 (Du et al., 2023). Additionally, the original half-hourly tower-based SIF data were temporally upscaled to daily and monthly values with the aid of PAR and NDVI, following the method described by Hu et al. (2018).

## 3 Results

### 3.1 Downscaled SIF dataset

The comparison of fine-resolution (0.05°) and coarse-resolution (1°) SIF datasets, derived from GOME, is illustrated in Fig. 2. The top two rows (panels a–f) illustrate the enhanced spatial variability achieved through the downscaling process, revealing finer vegetation patterns and distinct intensity gradients. The downscaled SIF datasets render subtle patterns in SIF more apparent compared to the original coarse-resolution data (panels g–k). Additionally, the downscaling method, which incorporates neighborhood-based pixel searching, effectively fills in data gaps in the original data while preserving spatial continuity. The residual, which was calculated as the difference between the downscaled SIF (which was re-aggregated to the original 1° resolution) and the original SIF, is shown in panels l–p. It can be observed that in major vegetated regions, the residuals are concentrated within the range of -0.50 to 0.50 mW m$^{-2}$ sr$^{-1}$ nm$^{-1}$. The histograms of the downscaling residuals across different years and sensors are shown in Fig. S2. Overall, the absolute values of the mean residuals are less than 0.008

250     mW m$^{-2}$ sr$^{-1}$ nm$^{-1}$, and the standard deviations are below 0.105 mW m$^{-2}$ sr$^{-1}$ nm$^{-1}$.

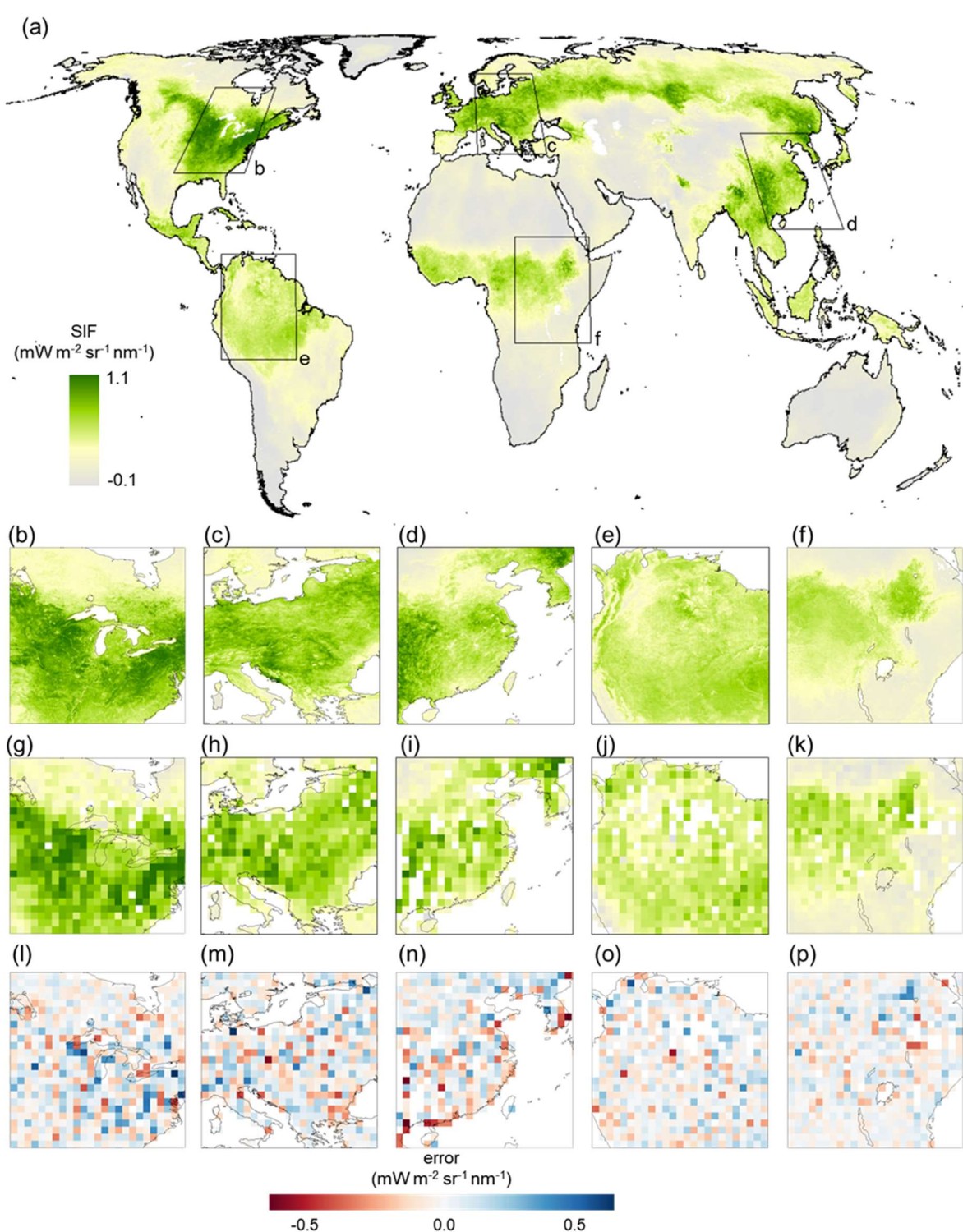

**Figure 2. Spatially downscaled SIF maps (a-f) compared to the original SIF maps at a coarser resolution (g–k). SIF data from GOME observations in July 1996 are shown as an example. The bottom row (l–p) shows the downscaling residuals, which were calculated**
**as the difference between the original SIF and the downscaled SIF, which was re-aggregated to the original resolution (1° × 1°). Panels b, g, and l depict North America; c, h, and m focus on Europe; d, i, and n depict East Asia (centered on China); e, j, and o represent the Amazon Basin; and f, k, and p show Sub-Saharan Africa.**

The distribution of monthly SIF before and after spatial downscaling is shown using GOME as an example (Fig. 3), while

results for the other three satellites are provided in Figs. S3–S5. The spatially downscaled SIF (0.05° × 0.05°) was re-

aggregated to 1° × 1° or 0.5° × 0.5° resolution for comparison with the original coarse-resolution SIF. The results demonstrate

that the SIF values from the re-aggregated pixels are generally consistent with the original SIF values, closely clustering along

the 1:1 line and showing strong agreement ($R^2 > 0.73$, RMSE $< 0.11$ mW m$^{-2}$ sr$^{-1}$ nm$^{-1}$), indicating that the LUE-based

downscaling model effectively captures the relationship between SIF and its driving variables.

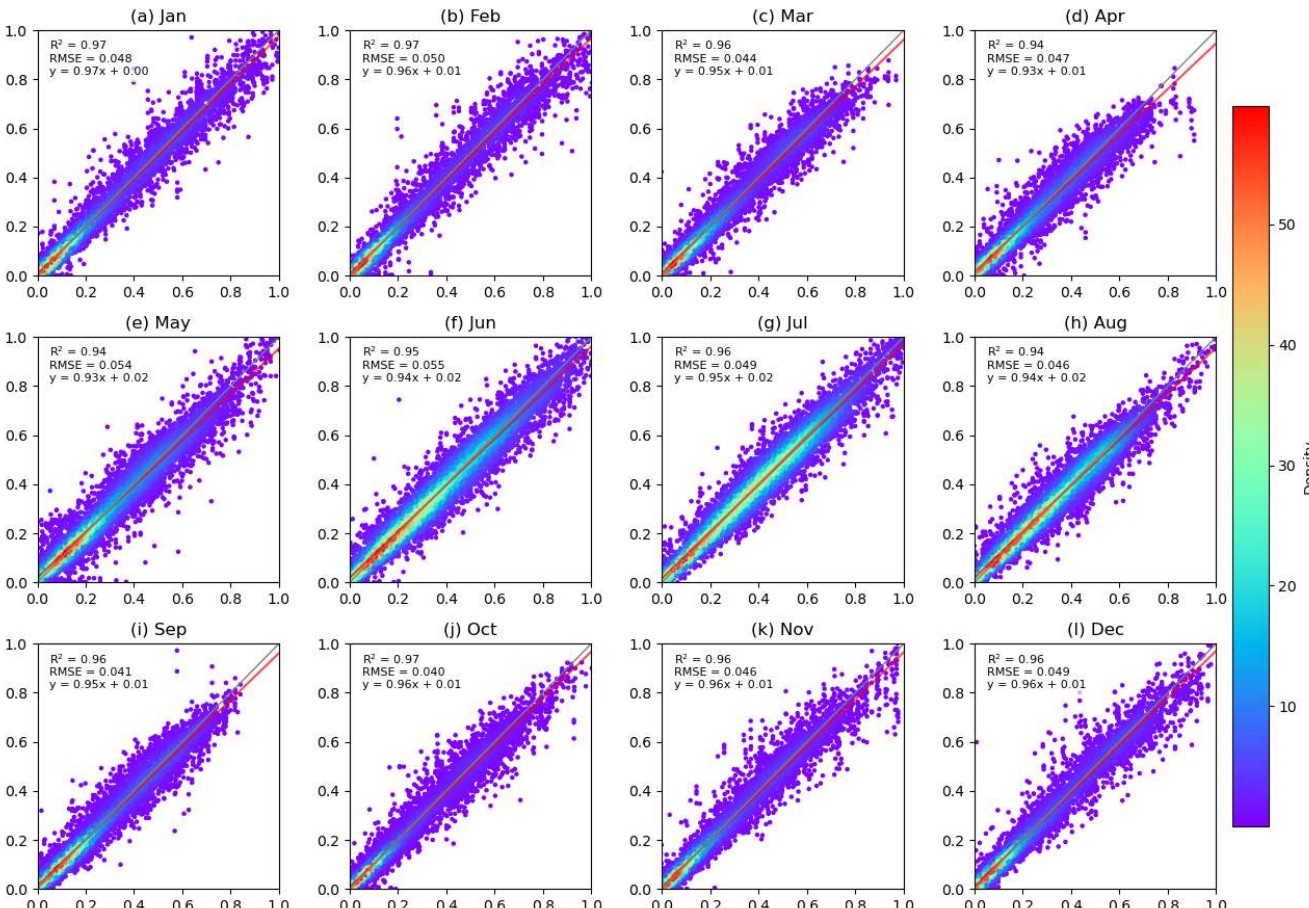

**Figure 3. The relationship between the reaggregated GOME SIF (SIF_reagg) and the original GOME SIF (SIF_original) for 1998 (by month).**

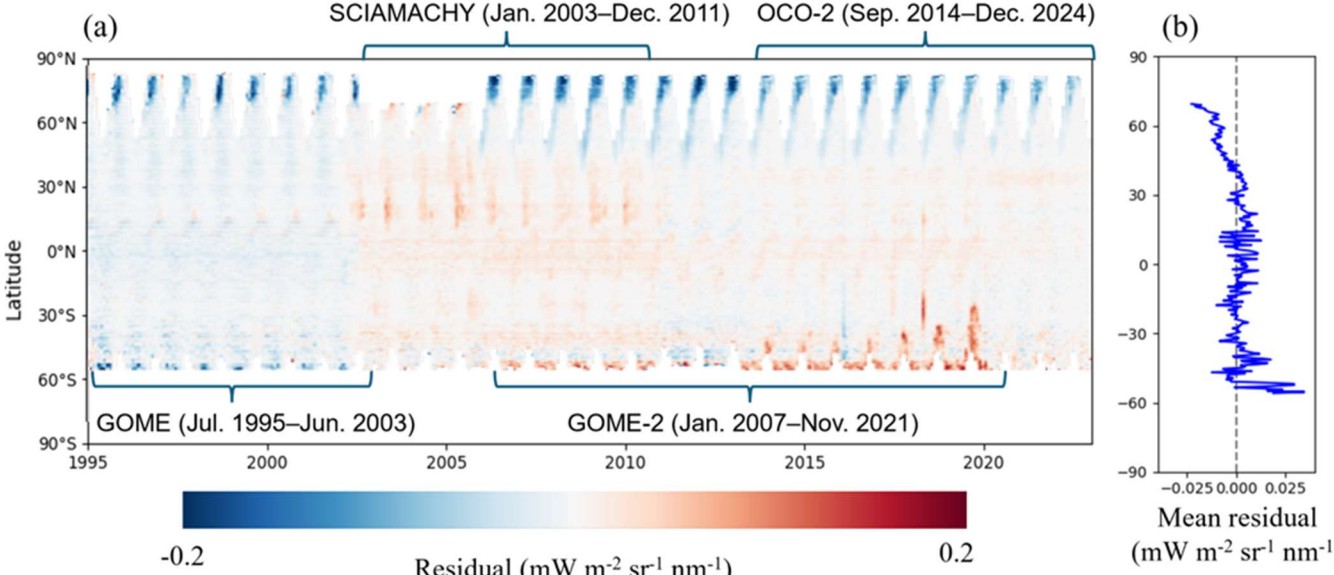

**Figure 4. The (a) time series and (b) temporal average of the latitudinally distributed residual generated by the LUE-based downscaling model. The residuals are calculated as the difference between the reaggregated SIF (SIF_reagg) and the original SIF (SIF_original). For consistency, the coarse spatial resolution SIF datasets were uniformly resampled to 0.5° × 0.5°. Only data for latitudes below 70° N are shown in (b).**

The temporal and spatial distributions of the spatial downscaling residuals were analyzed (Fig. 4). The monthly mean residuals across different latitudes and months were generally below 0.2 mW m$^{-2}$ sr$^{-1}$ nm$^{-1}$ (Fig. 4 a). In addition, the regions with relatively larger residuals (e.g., > 0.1 mW m$^{-2}$ sr$^{-1}$ nm$^{-1}$) were mainly located in high-latitude areas. As shown by the temporally averaged residuals (Fig. 4 b), for most areas below 70°N, the absolute mean residuals are less than 0.05 mW m$^{-2}$ sr$^{-1}$ nm$^{-1}$ for regions below 70° N. These results indicate that the downscaling method maintains high consistency with the original data across a broad range of temporal and spatial conditions.

### 3.2 Temporal harmonization

The time series of the original SIF datasets from individual satellites and the resulting long-term harmonized SIF dataset (1995–2024) are presented in Fig. 5. Before normalization, substantial inter-sensor discrepancies were observed: mean SIF values ranged from 0.19 mW m$^{-2}$ sr$^{-1}$ nm$^{-1}$ (SCIAMACHY) to 0.28 mW m$^{-2}$ sr$^{-1}$ nm$^{-1}$ (GOME), while interannual trends varied from -0.76% yr$^{-1}$ (GOME) to 0.54% yr$^{-1}$ (GOME-2). Among the original sensor datasets, only the GOME-2 dataset showed a statistically significant trend (p < 0.05), whereas other sensors exhibited non-significant variations (p ≥ 0.05). In contrast, the harmonized LHSIF dataset demonstrated a significant positive trend (p < 0.001, Trend = 0.31% yr$^{-1}$).

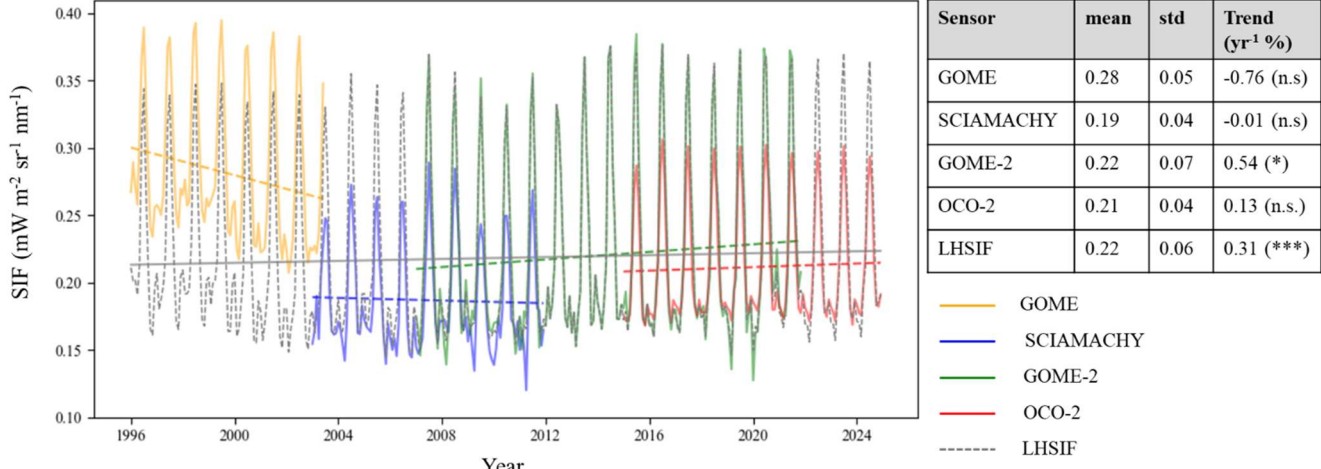

**Figure 5. Global-averaged SIF time series derived from GOME (yellow), SCIAMACHY (blue), GOME-2 (green), and OCO-2 (red), along with the long-term harmonized SIF time series (LHSIF, gray dotted line), which aligns the satellite datasets based on overlapping periods. The table on the right summarizes the statistical characteristics of each sensor, including the mean, standard deviation (std), and the annual trend (Trend) averaged over the respective periods. The statistical significance of the trends is indicated as follows: n.s. for not significant (p > 0.05), * for significant (p < 0.05), and *** for highly significant (p < 0.001).**

Error analyses were conducted for different climatic zones and plant functional types. Fig. 6 shows the comparison between GOME-2 and SCIAMACHY SIF, both before and after normalization. In all tested scenarios, the normalization process substantially reduced the differences between the two sensors. Overall, the MSD decreased by more than 49% following normalization. In most cases, the difference in the average (bias, shown in red) was the dominant component of the MSD between GOME-2 and SCIAMACHY SIF before normalization. In the temperate and tropical zones of the Southern Hemisphere, discrepancies were primarily attributed to variations in variance (shown in green) and weak correlations (shown in blue). The MSD was significantly reduced after temporal correction, with a notable decrease in the proportion of bias. Only a small proportion (~0.002 mW m$^{-2}$ sr$^{-1}$ nm$^{-1}$) of phase-related errors remained in the corrected dataset.

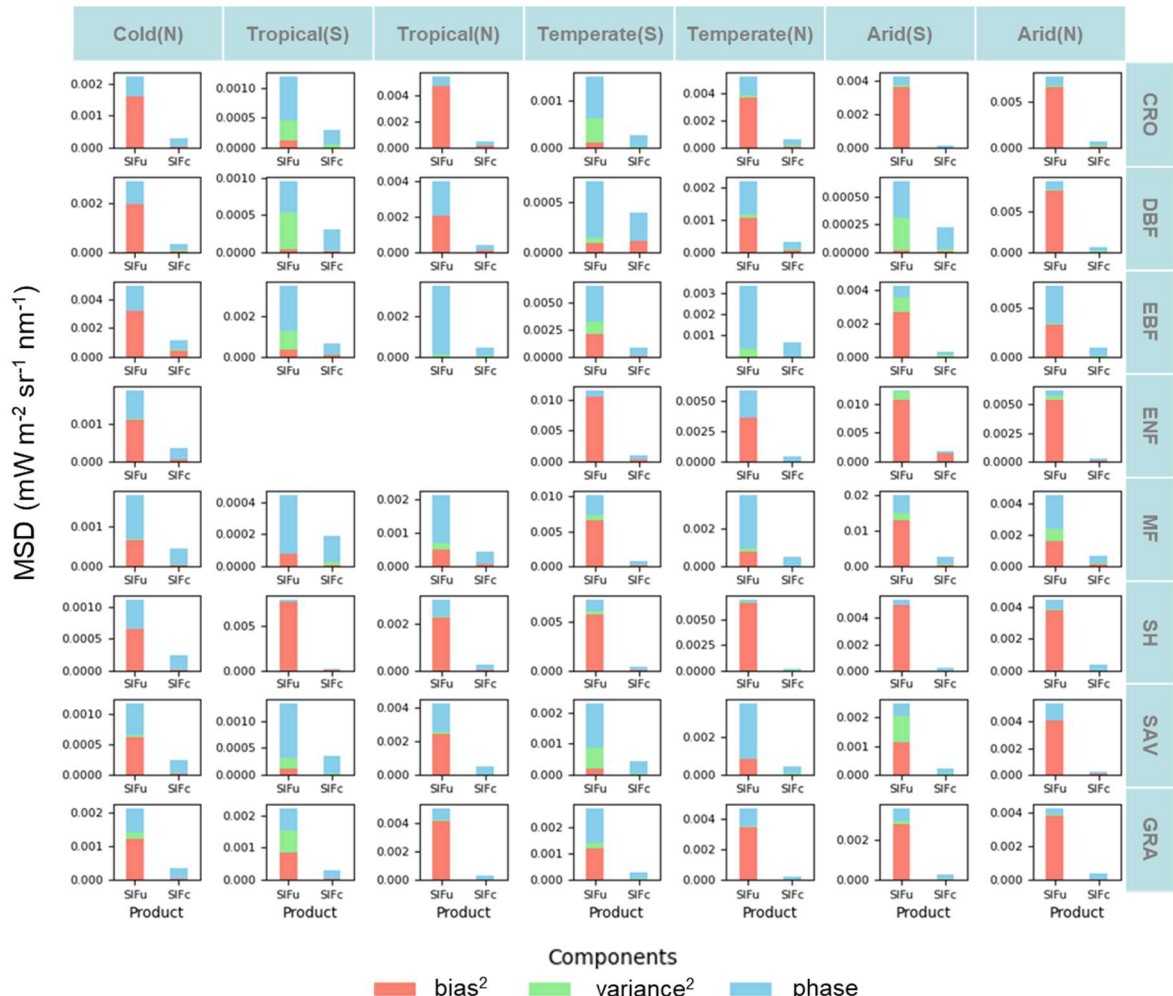

Figure 6. The mean squared difference (MSD) between GOME-2 SIF and SCIAMACHY SIF before ($SIF_u$) and after ($SIF_c$) normalization. The results show the average conditions across different climatic zones and vegetation functional categories during 2007.

The annual maximums of the global-averaged SIF were used to investigate the fluctuation of the worldwide vegetation from 1995 to 2024. Significant interannual fluctuations were found for the SIF time series without normalization, with an overall decline (blue line in Fig. 7a). The normalized SIF time series reveals a growth rate of 0.31% yr$^{-1}$. After normalization, the standard deviation of the fitted slope decreases from 0.25% to 0.07%, indicating a reduction in uncertainty. The boxplot in Fig. 7b further shows a narrower range of SIF values after temporal normalization, suggesting a more concentrated data distribution and improved comparability across sensors.

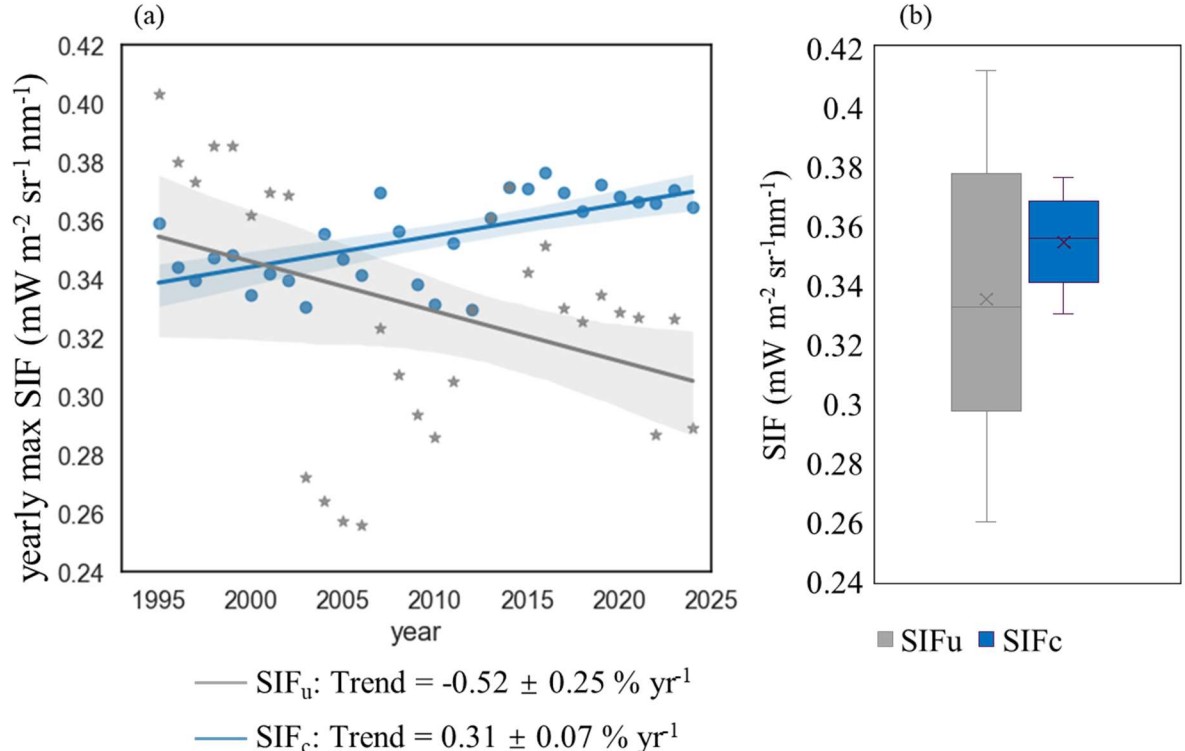

**Figure 7. (a) Trend and (b) box plot of the yearly maximum global-averaged SIF of the combined time series before (SIFu) and after (SIFc) normalization during 1995–2024.**

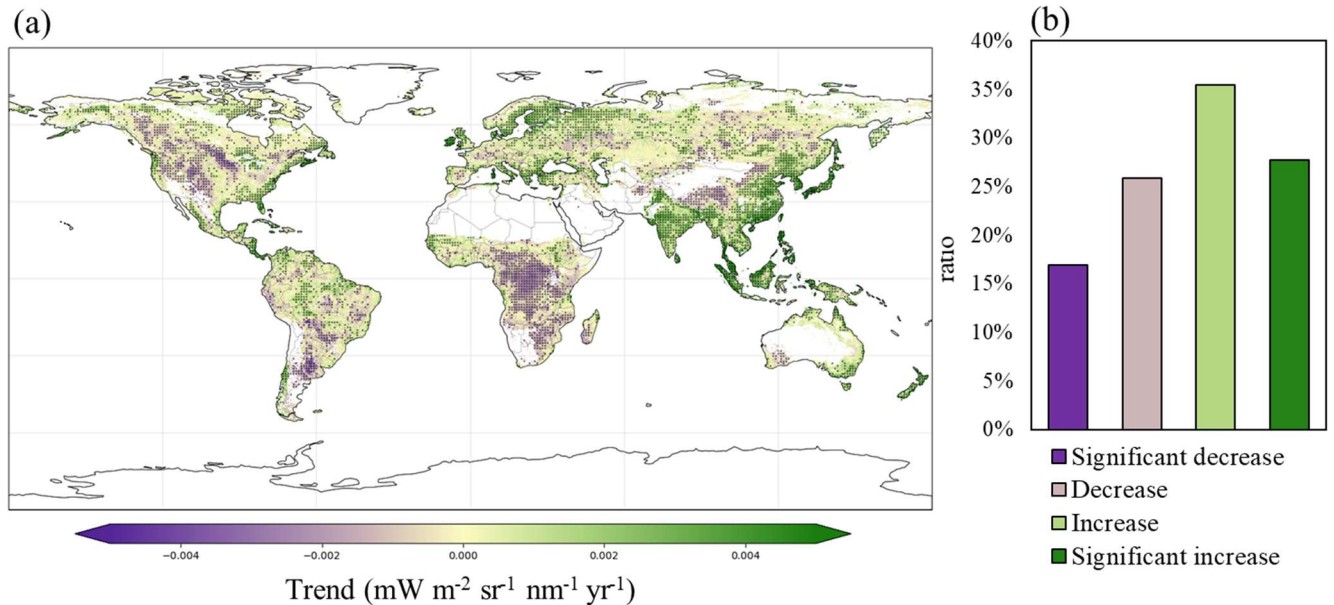

**Figure 8. (a) Map of trends in LHSIF for 1996–2024. (b) Percentage of areas in global vegetation covered by four different trend types (significant decrease: negative correlation and $p < 0.05$; decrease: negative correlation and $p \geq 0.05$; increase: positive correlation and $p \geq 0.05$; and significant increase: positive correlation and $p < 0.05$). The black dots in (a) represent statistically significant trends ($p < 0.05$). The statistics begin in 1996 due to incomplete data coverage in 1995, which only includes the second half of the year (July–December).**

Significant SIF increases are observed in South and Southeast Asia, as well as parts of Eastern Europe. Conversely, significant declines in SIF are mainly found in Southern Africa and parts of Western North America. In Australia, the eastern regions show slight increases in SIF, while the western regions experience declines. Overall, SIF growth occurred in about 63% of the world's

vegetated areas, with significant increases observed in around 28% of these areas between 1996 and 2024 (Fig. 8b).

## 3.3 Validation and comparative analysis

The annual maximum LHSIF exhibited a positive trend (0.31 ± 0.07% yr$^{-1}$), with data points clustering around the trendline (Fig. 9a). The growth rate of LHSIF (0.31% yr$^{-1}$) closely aligns with that of BEPS GPP (Fig. 9f, 0.47% yr$^{-1}$), demonstrating LHSIF's stability in capturing long-term trends of GPP. LT_SIFc* also shows a positive trend but with a lower growth rate (Fig. 9b). The SIF_005 product exhibits a negative trend during 2003–2017 in stark contrast to all other datasets (Fig. 9c). Although the spurious trends have been largely corrected for the original SIF products used by SIF_005, the long-term trend remains suboptimal.

From 1995 to 2021, less pronounced trends were shown by AVHRR NDVI (Fig. 9d, 0.18 ± 0.02% yr$^{-1}$) and NIRv (Fig. 9e, 0.34 ± 0.02% yr$^{-1}$) compared with SIF and GPP. Compared to SIF-based products, NDVI is more susceptible to interference from vegetation canopy structure and non-photosynthetic processes; thus, it is less effective at capturing photosynthetic activity. In this regard, LHSIF provides a more direct indication of photosynthesis and can supplement NDVI and NIRv in detecting changes in GPP.

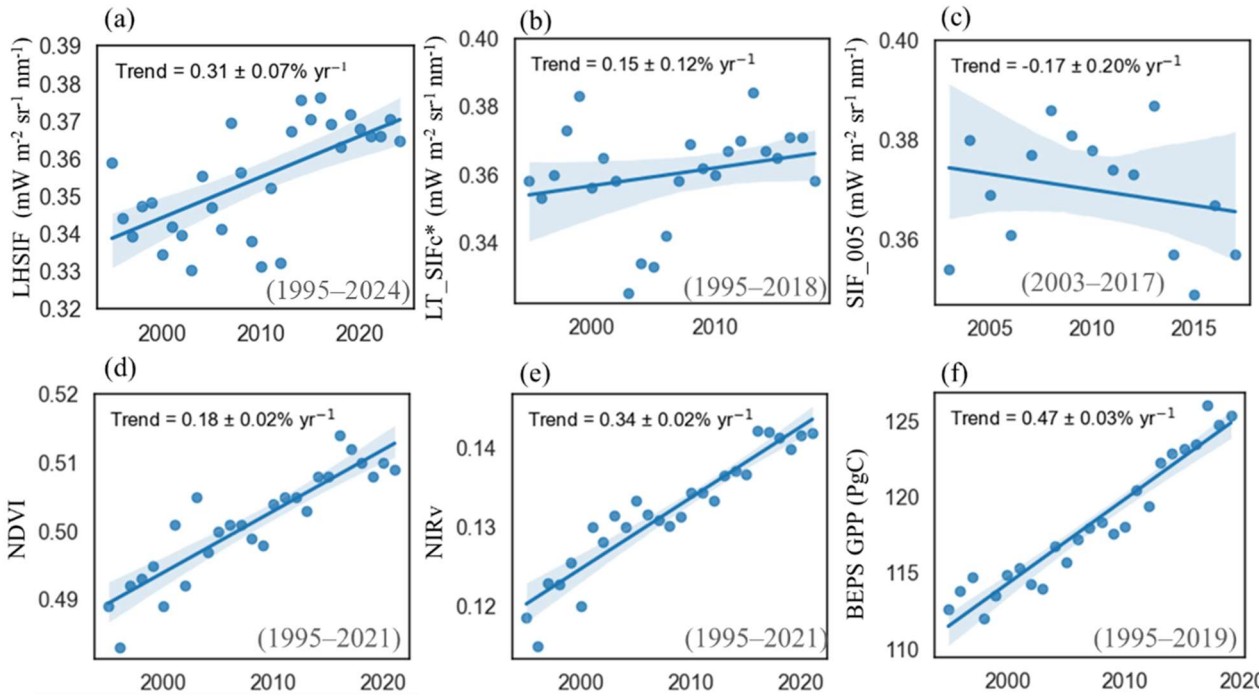

**Figure 9. The yearly maximum (a) LHSIF compared with (b) LT_SIFc*, (c) SIF_005, (d) AVHRR NDVI, (e) AVHRR NIRv, as well as the annual total (f) BEPS GPP.**

The interannual trends of several long-term SIF products—including LHSIF, LT_SIFc*, SIF_005, and LCSIF—were compared. The annual maximum of global monthly SIF was used for comparison. Fig. 10 presents the results for the global scale as well as for several representative climate zones and land cover types.

Among the four SIF products, all except SIF_005 show increasing global trends. LHSIF exhibits the strongest upward trend

at 0.31% yr$^{-1}$, while LCSIF presents the most stable interannual variation, with a trend standard deviation of only 0.01% yr$^{-1}$. LHSIF and LCSIF display statistically significant increases on the global scale, whereas the trends for LT_SIFc* and SIF_005 are not statistically significant. The divergent trend between SIF_005 and the other SIF products is further demonstrated on regional scales. For example, in continental cropland regions (Fig. 10d) and temperate deciduous broadleaf forest (DBF) biomes, LHSIF, LT_SIFc*, and LCSIF generally exhibit consistent positive trends, whereas SIF_005 shows a declining trend.

In most cases shown in Fig. 10, LHSIF, LT_SIFc*, and LCSIF display consistent trends. An exception occurs in arid regions, where LCSIF shows an increasing trend while both LHSIF and LT_SIFc* exhibit decreasing trends (Fig. 10j,k). This divergence may be attributed to the machine learning–based nature of LCSIF, which relies heavily on predictor variables and may not fully capture the actual SIF dynamics under stress conditions. In contrast, the observational basis of LHSIF enables it to more directly reflect regional responses to environmental variability.

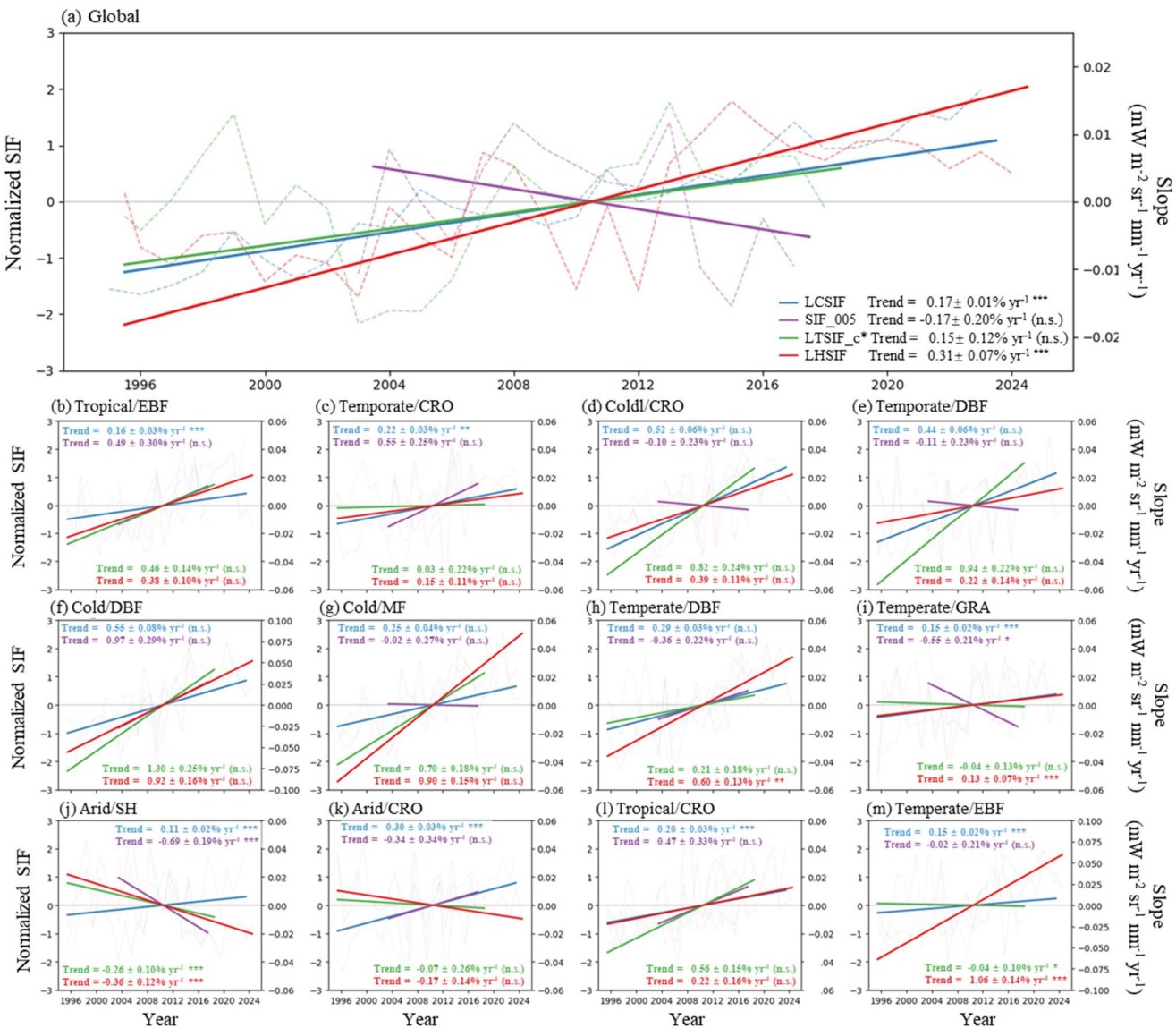

Figure 10. Comparison of interannual variations in long-term SIF products. LHSIF (red), LT_SIFc* (green), SIF_005 (purple), and LCSIF (blue) are compared for (a) the global scale and (b–m) various climatic and vegetation regions. All datasets were normalized

using the z-score method. Dashed lines represent yearly maximum values, and solid lines indicate linear trends. To aid visual comparison, trend lines were anchored at the origin (2010, 0). The statistical significance of the trends is indicated as follows: n.s. for not significant (p > 0.05), * for significant (p < 0.05), and *** for highly significant (p < 0.001).

In addition to long-term satellite products, ground-based observations were also incorporated for comparison. The relationships of LHSIF and AVHRR NDVI with FLUXNET GPP are illustrated in Fig. 11. The LHSIF product shows a strong ability to track GPP, especially for cropland and mixed forest types (Fig. 11a). In contrast, NDVI consistently exhibits lower $R^2$ values (Fig. 11b) and a more pronounced nonlinear relationship with GPP due to saturation effects. Apart from a few groups in the Southern Hemisphere (such as grasslands in tropical and arid areas), where only a small number of sites are available (see Fig.

S1), SIF outperforms NDVI in most cases.

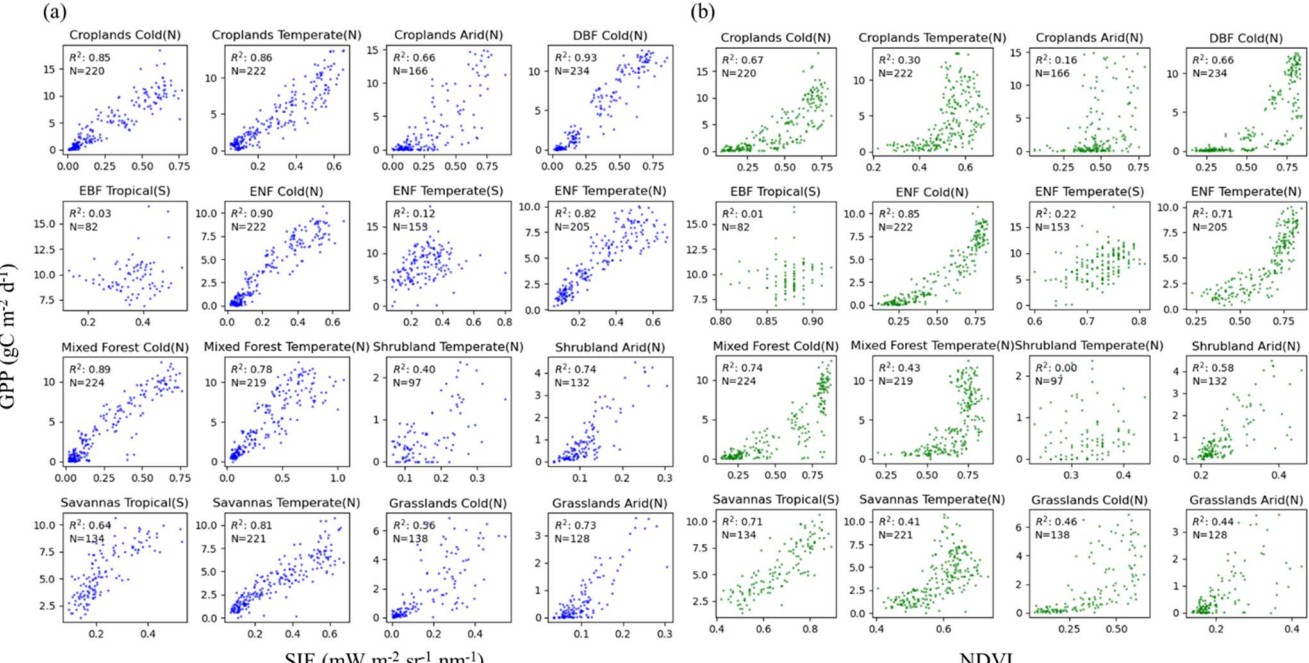

**Figure 11. Comparison of long-term relationships between SIF vs. GPP and NDVI vs. GPP at FLUXNET sites. Only those flux tower sites that have accumulated over a decade of data were chosen and subsequently categorized according to their respective climate zones and vegetation types.**

Additionally, comparisons were conducted between LHSIF and the tower-based SIF measurements at five ChinaSpec sites. As a result, LHSIF demonstrated strong agreement with tower-based SIF measurements both temporally and spatially (Fig. 12). The consistency of the intra-annual variations was evident between LHSIF and the in-situ measurements for each site. As shown in the right panel, the monthly composite values are highly correlated, with most points clustering near the 1:1 line and correlation coefficients generally exceeding 0.6.

However, some deviations were observed. For example, at the Gucheng (GC) and Xiaotangshan (XTS) sites, which are characterized as wheat-maize rotation croplands, discrepancies occurred in June. During this month, tower-based SIF measurements recorded a trough when wheat was harvested and maize had yet to emerge. Due to spatial heterogeneity, LHSIF

was unable to capture this phenomenon, resulting in a reduced correlation between LHSIF and in-situ measurements at these

two sites. To highlight the overall correlation, the data in June for these two sites were removed from the scatter plot (Fig. 12h, j).

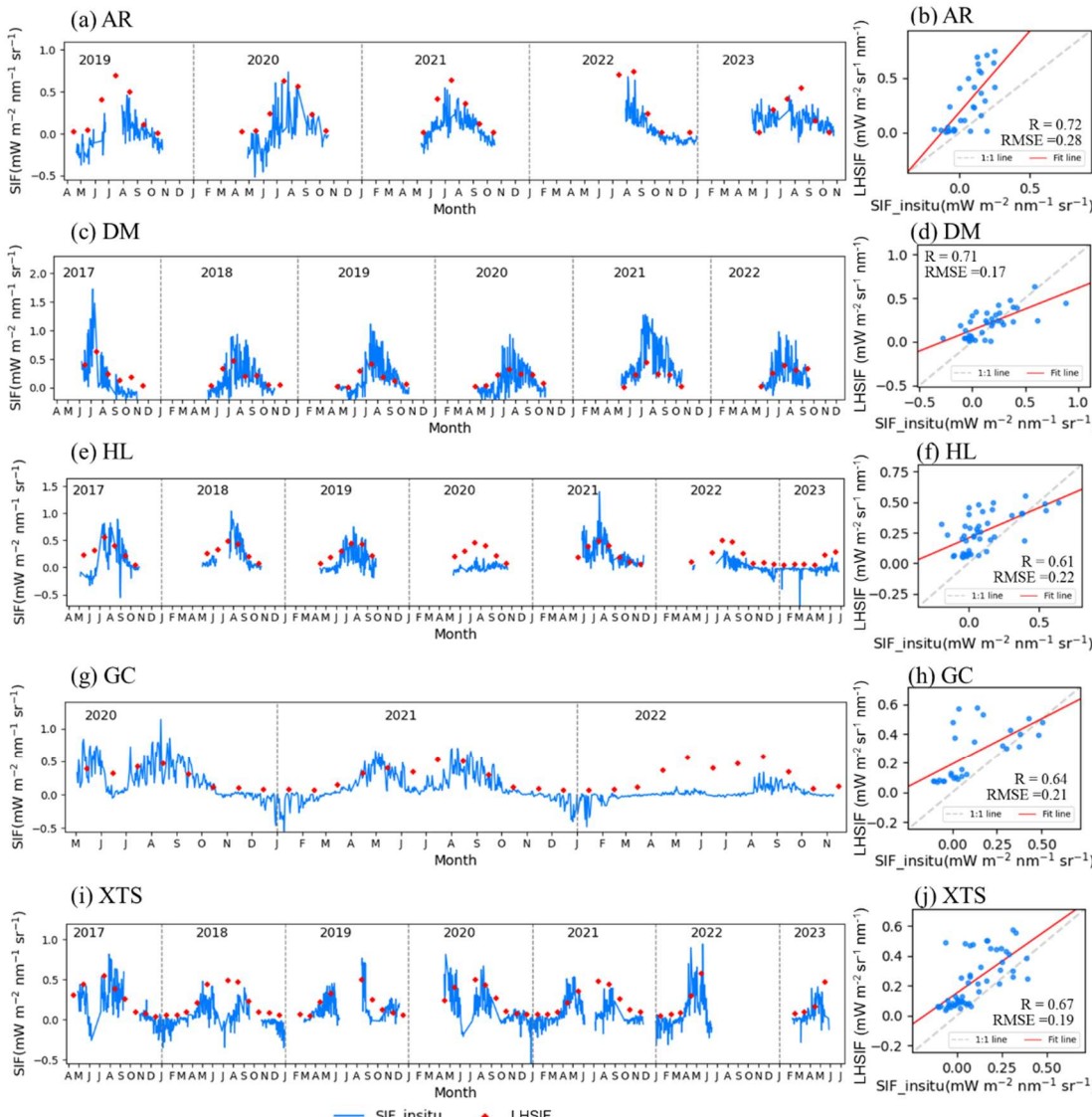

**Figure 12. Comparison between LHSIF with the tower-based SIF measurements (SIF_insitu) at (a, b) AR, (c, d) DM, (e, f) HL, (g, h) GC, and (i, j) XTS sites. The temporal pattern of LHSIF compared with daily and monthly averaged SIF_insitu is shown in the left**
**and right columns, respectively. The horizontal axis of the chart in the left column represents the first letters of the month names. Data points in June were excluded from the scatter plot for GC and XTS (h, j).**

## 4    Discussion

### 4.1 Improvements in cross-sensor harmonization

In this study, we applied a CDF normalization method to harmonize cross-sensor SIF measurements. While the general concept
and algorithm are similar to previous studies (Wen et al., 2020; Wang et al., 2022), our data processing framework incorporates
several key improvements.

Firstly, a temporally corrected GOME-2A SIF (TCSIF) dataset (Zou et al., 2024) was used as the reference baseline. The TCSIF product incorporates radiometric correction of GOME-2A sensor degradation using a pseudo-invariant method and underwent a two-step validation at both radiance and SIF levels. As shown in Zou et al. (2024), the interannual variability of TCSIF shows strong consistency with GPP, providing a more robust reference for long-term harmonization. In contrast, the SIF_005 dataset (Wen et al., 2020), which was based on the original GOME-2 SIF, shows pronounced interannual fluctuations and a declining trend over 2003–2017, likely due to residual degradation effects (see Fig. 9c and Fig. 10).

Secondly, our harmonization strategy uses GOME-2A as the reference sensor. Its extended data record (2007–2021) provides over five years of overlap with both SCIAMACHY and OCO-2, allowing us to perform a single-step normalization for each. This approach helps reduce the uncertainty propagation associated with multi-step corrections. In contrast, the LT_SIFc* product uses GOME as the benchmark, relying on only a six-month overlap with SCIAMACHY and then sequentially calibrating SCIAMACHY and GOME-2A, which may accumulate uncertainties.

To quantify the impact of overlap duration on harmonization uncertainty, we performed normalization experiments using 6-, 12-, and 24-month overlap periods between GOME-2 and SCIAMACHY/OCO-2 (Fig. 13). Each experiment was repeated 10 times to assess the variability of the resulting harmonized time series.

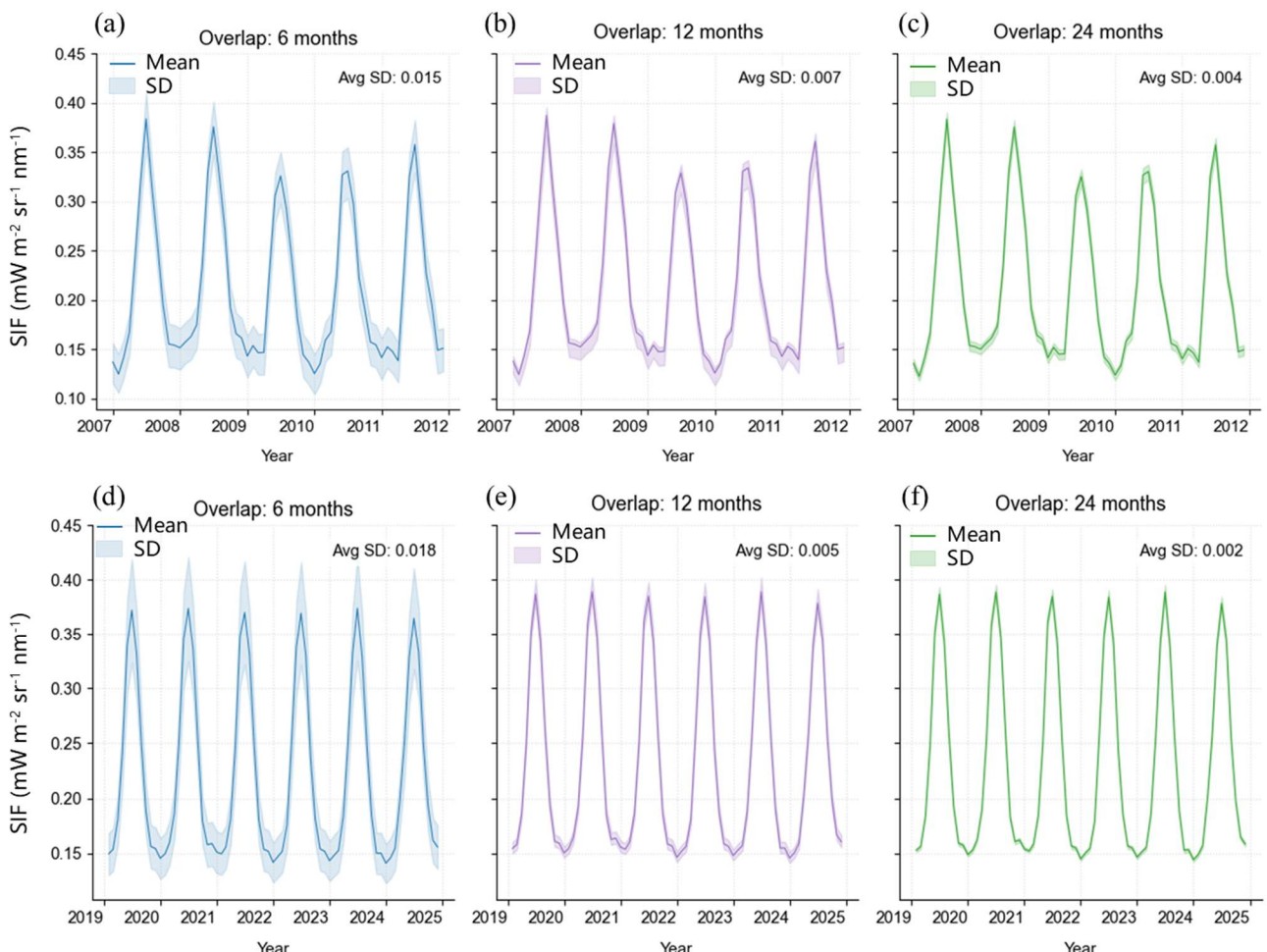

**Figure 13 Normalized SIF time series using different overlap durations: 6 months (first column), 12 months (second column), and 24 months (third column) from SCIAMACHY (top row) and OCO-2 (bottom row). The shaded areas represent the standard deviation (SD) across multiple experiments, with units of mW m⁻² sr⁻¹ nm⁻¹.**

The results show that a six-month overlap leads to a higher standard deviation in SIF time series compared to longer overlaps. As the overlap period  was extended from 6 to 12 months, the standard deviations of the normalized SIF series decreased from 0.015 to 0.007 mW m$^{-2}$ sr$^{-1}$ nm$^{-1}$ (SCIAMACHY) and from 0.018 to 0.005 mW m$^{-2}$ sr$^{-1}$ nm$^{-1}$ (OCO-2), representing a reduction of over 53.3%. These results confirm that short overlap periods increase normalization uncertainty and highlight the robustness of our chosen strategy, which avoids using GOME as the baseline. Nevertheless, users should be aware of the potential uncertainties in the early portion (from 1995 to 2003) of the LHSIF record.

Furthermore, in contrast to the pixel-by-pixel matching methods adopted in previous studies, we applied a region-based and month-specific normalization strategy. The region-based approach allows for a larger sample size within each region, potentially enabling a more robust estimation of the CDF, while the month-specific treatment helps account for seasonal variations in the CDF. The improved stability of interannual trends in the LHSIF product, compared to LT_SIFc* and SIF_005 at both global and regional scales (Fig. 10), appears to reflect the effects of this normalization strategy.

**4.2  Limitations and future perspectives**

Our investigation shows that the CDF normalization approach effectively reduces disparities across sensors, providing a unified reference framework with the longest time series to date. Although the normalized series yields consistent seasonal and interannual trends across sensors, it is important to note that this normalization approach does not involve a physically based calibration (e.g., radiometric calibration based on pseudo-invariant targets). As a result, residual sensor-specific biases may still exist. The most robust approach for cross-sensor calibration is based on pseudo-invariant calibration sites (PICs) located in non-vegetated areas (Markham and Helder, 2012; Khakurel et al., 2021). This method has been successfully applied to the normalization and long-term monitoring of reflectance data and vegetation index products (Angal et al., 2013; Mishra et al., 2014; Jeong et al., 2024; Tavora et al., 2023). However, in commonly used PICs, such as deserts and water surfaces, SIF signals are inherently weak and highly susceptible to noise that can cause significant uncertainty in PIC-based calibration for SIF applications.

Additionally, our downscaling approach follows the methodology proposed by Duveiller et al. (2016, 2020), where NIRv is used in the LUE model instead of NDVI to enhance the model's interpretability for SIF (Badgley et al., 2017). Nevertheless, some explanatory variables remain unaccounted for. For instance, incorporating PAR could improve model interpretability under cloudy conditions (Ryu et al., 2018). However, discrepancies in overpass time and scale effects can cause inconsistencies between PAR and SIF products, which may increase uncertainties in the downscaling model. Furthermore, incorporating fluorescence escape efficiency (Ryu et al., 2019) and topographic factors (Chen et al., 2022; Tao et al., 2024) into the downscaling model could further enhance its performance.

Model selection is ultimately more critical than the choice of input variables (Duveiller et al., 2020). Previous research on spatial downscaling was predominantly using purely empirical machine-learning approaches (Gentine and Alemohammad, 2018; Wen et al., 2020; Hong et al., 2022; Lu et al., 2024). An alternative strategy redistributes the initial global downscaling results based on the original coarse-resolution SIF values and retains the characteristics of the original observational signal to the greatest extent possible (Ma et al., 2022; Chen et al., 2025). Our experimental results confirm that our downscaled SIF products also remain consistent with the original signals (Fig. 4). In addition, the LUE-based approach incorporates physiological constraints, ensuring that the downscaled SIF values remain within a reasonable range compared to traditional machine-learning models.

Another type of long-term SIF datasets have been generated by temporally extrapolating SIF observations based on machine-learning methods. These datasets provide more than two decades of high-temporal-resolution data beyond the monthly scale (Zhang et al., 2018b; Li and Xiao, 2019). However, such datasets predominantly depend on model-driven predictions

constrained by satellite observation periods, rather than being based on actual observational data, which is fundamentally different from the approach we employed. Currently, the temporal resolution of purely observation-based enhanced SIF products that span longer than 20 years remains constrained at the monthly scale, largely due to noise in the satellite SIF products. Overcoming this limitation will require further refinement of existing downscaling models, paving the way for future products to achieve a resolution of 16 days or higher.

## 5    Conclusion

In this study, we developed a long-term harmonized SIF dataset (LHSIF) spanning 1995 to 2024. SIF datasets from various satellites were normalized using multi-sensor SIF observations through a CDF normalization approach, using the temporally corrected GOME-2A SIF dataset as a benchmark. An LUE-based model was used for spatial downscaling, yielding a fine resolution of 0.05° with an absolute mean residual less than 0.05 mW m$^{-2}$ sr$^{-1}$ nm$^{-1}$.

Our analysis demonstrated that the harmonized dataset significantly reduced overall errors by more than 49% and exhibited a stable interannual increase ($0.31 \pm 0.07\%$ yr$^{-1}$). The interannual trend of LHSIF closely aligns with the growth of GPP ($0.47 \pm 0.03\%$ yr$^{-1}$) and demonstrates superior temporal and spatial consistency compared to NDVI. Validation against ground-based SIF observations ($R > 0.6$) further underscores the reliability of the harmonization approach and the dataset's utility in global vegetation studies.

By focusing on the harmonization of satellite-derived SIF products, the LHSIF dataset offers a unified framework for integrating multi-sensor SIF data to enable long-term monitoring of global photosynthesis. This contribution provides an essential tool for understanding vegetation responses to environmental changes and advancing the field of Earth system science.

## 6    Data availability statement

The LHSIF dataset generated in this study is publicly available at https://doi.org/10.5281/zenodo.16394372 (Zou et al., 2025). Additional information regarding the data and methods is available upon request from the corresponding author.

## 7    Author contribution

CZ and LL designed the experiments, CZ carried them out. CZ, SD and XL developed the model code and generated the products. CZ prepared the manuscript with contributions from all co-workers.

## 8 Competing interests

The contact author has declared that none of the authors has any competing interests.

## 9 Acknowledgments

This work was supported by the National Natural Science Foundation of China (grant number: 42425001).

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
