# Peer review of "Development of the Long-term Harmonized multi-satellite SIF (LHSIF) dataset at 0.05° resolution (1995–2024)"

_Earth System Science Data, 2025_

## Referee Comment (RC1)

This paper produced a long-term SIF dataset (LHSIF) from 1995 to 2023. This topic is of significant importance for global-scale carbon simulation and vegetation studies. I think the datasets preprocessing procedures are solid. However, there are some questions I am concerned, especially about the reliability of the inter-annual trend in the SIF data from 1995 to 2000. There are many spelling errors in sentences throughout the paper. I listed as much as I could found. Please double check to avoid such errors and also try to avoid using 'Figure.... illustrated...' as the begining of a paragraph in the result section. Following a logical writing orders is necessary. I rcommend minor revision and here are some comments I wish the authors could give explanations before accepting this paper for publication.

**Minor Reviews:**

1) You didn't mention how do you do quality control for GPP provides by FLUXNET. Did you use the quality flag data of "GPP_DT_VUT_REF" during SIF production?

2) I am concerned whether the interannual trend in SIF before 2000 is true or reliable. For example, in Figure 7, we did not see much difference in yearly maximum global-averaged SIF based on the SIFu or SIFc from 2000 to 2023. The large discrepency mainly occurs in year 1995 to 2000, when SIFu is significant higher than the SIFc with a significnatly decreasing trend (mainly provided by GOME?). However, there exists larege discrenpency between SIFu and SIFc before 2000. How could you convience the readers that such inter-annual trend in SIF in year 1995 to 2000 is reliable? There is no validation of this trend from 1995 to 2000 as far as I can see in the paper.

3) Line 21: You should better change a word to substitute 'tool' in line 21. How could a dataset be a tool? How about "Therefore, the long-term harmonized SIF dataset with a fine 0.05° resolution is valuable for estimating global photosynthesis over extended periods."
4) Missing space in multiple places in sentences.
    Line 22: Missing space: "The LHSIF dataset is available at https://doi.org/10.5281/zenodo.14854185 (Zou et al., 2025)."
    Line 138: There should be a comma in "0.5 ° × 0.5 °, and 1 ° × 1 ° ".
    Line 144: GPP (Berry et al)
    Line 153: Missing space : 2 m.
    Line 154: Missing comma: f(NIRv), f(VPD), and f(AT),
    Line 159: "L-BFGS-B" algorithm (Byrd et al., 1995)
    Line 167: Two balnk spaces here. "0.05° , were"
    Line 445: Missing space before the reference.
5) Incomplete sentence in Line 267: "The temporal and spatial distributions of the spatial downscaling residuals were analyzed (Fig. 4). The residual was calculated as the difference between ? As shown..."
6) L353: Please do grammer check and change this sentence to: "To highlight the overall

correlation, the data in June for these two sites were removed from the scatter plots."

7) Line 434: Upper foot label for m-2 in "residual less than 0.05 mW m$^{-2}$ sr$^{-1}$ nm$^{-1}$".

8) Line 303: please add an 'and' in "0.05; and significant increase".

---

## Author Comment (AC1)

https://doi.org/10.5194/essd-2025-94
**Response to Reviewer 2 Comments:**

The author provides a systematic work for the development of the Long-term Harmonized multi-satellite SIF (LHSIF) dataset by coordinating SIF satellite observations from GOME, SCIAMACHY, GOME-2, and OCO-2. The TCSIF dataset, also developed by the same author, provides strong support for this study. The overall technical process is complete and feasible, and some interesting results are obtained. This study fits well within the scope of ESSD journal. However, I have some comments about the development process of this dataset. Therefore, my recommendation is major revision.

We sincerely appreciate the reviewer's thoughtful feedback and recognition of our work. In response to the comments, we have implemented the following key improvements to both the dataset and manuscript:

1. **Enhanced harmonization method**

   We replaced the moment matching approach with a CDF-based harmonization strategy, where cumulative distribution functions were constructed separately for different vegetation-climate zone combinations. To preserve seasonal dynamics, the matching was performed independently for each calendar month.

2. **Extended temporal coverage**

   The dataset now includes updated records through 2024.

3. **Strengthened Validation**

   We have strengthened the validation section, incorporated more quantitative evaluations, and revised or removed subjective language to enhance the scientific clarity and objectivity of the manuscript.

Detailed responses to each comment are provided below. Reviewer comments are shown in black, authors' responses in blue, and the corresponding revisions to the manuscript in purple.

Specific concerns and suggestions are outlined as follows:

**Major Comment 1:** I have concerns regarding the innovation and validation of the dataset. The authors mention two sets of long-term harmonized SIF products (Wang and Wen) and briefly describe their processing steps. However, neither in the Introduction nor in the Results/Discussion sections do I clearly see what specific methodological innovations the authors have introduced or where their data demonstrate superior performance. For instance, while they argue that the CDF method has limitations, there appears to be insufficient validation and justification—such as a comprehensive comparison of overall errors across the three datasets or scenario-specific analyses. I recommend that the authors conduct additional analyses to provide more quantitative evidence supporting the advantages of their dataset.

**Response:** We sincerely thank the reviewer for this valuable comment. First, we would like to clarify that the main objective of this study is not to propose a new harmonization algorithm, but rather to develop a long-term SIF dataset based solely on satellite observations, with improved consistency and temporal coverage.

In the original submission, we explored moment-matching as an alternative to the CDF method. This consideration was motivated by the inherent limitations of pixel-level CDF normalization, as adopted in previous studies (e.g., Wen et al., 2020; Wang et al., 2022). Specifically, the limited number of overlapping observations at the pixel scale can make it more susceptible to temporal noise, potentially reducing the reliability of the matching.

To address this issue, the revised version adopts an ecological zone-based normalization approach. By grouping pixels into broader ecological regions (e.g., Köppen climate zones), we increased the sample size for each normalization unit, enabling a more reliable implementation of the CDF matching method. As a result, the revised version retains the CDF method. Although the method remains consistent with prior studies, this adjustment reduces the potential instability caused by limited overlapping data in short time series.

We updated the methodology section (Section 2.3) to detail this adjustment and added quantitative assessments in the results section to demonstrate the behavior of the harmonized dataset under this framework.

**2.3 CDF matching method**

[revised manuscript text omitted]

Figure 1. Comparison of interannual variations in long-term SIF products. LHSIF (red), LT_SIFc* (green), SIF_005 (purple), and LCSIF (blue) are compared for (a) the global scale and (b–m) various climatic and vegetation regions. All datasets were normalized using the z-score method. Dashed lines represent yearly maximum values, and solid lines indicate linear trends. To aid visual comparison, trend lines were anchored at the origin (2010, 0). The statistical significance of the trends is indicated as follows: n.s. for not significant (p > 0.05), * for significant (p < 0.05), and *** for highly significant (p < 0.001).

We have added the following paragraph to the revised discussion section to summarize the improvements of our dataset:

**4.1 Improvements in cross-sensor harmonization**

In this study, we applied a CDF normalization method to harmonize cross-sensor SIF measurements. While the general concept and algorithm are similar to previous studies (Wen et al., 2020; Wang et al., 2022), our data processing framework incorporates several key improvements.

Firstly, the harmonization process is anchored on a temporally corrected GOME-2A SIF (TCSIF) dataset (Zou et al., 2024), which accounts for long-term sensor degradation. The TCSIF product incorporates radiometric correction of GOME-2A sensor degradation using a pseudo-invariant method, and underwent a two-step validation at both radiance and SIF levels. As shown in Zou et al. (2024), the interannual variability of TCSIF shows strong consistency with GPP, providing a more robust reference for long-term harmonization. In contrast, the original GOME-2 SIF used in SIF_005 (Wen et al., 2020) exhibits substantial interannual

https://doi.org/10.5194/essd-2025-94

fluctuation and a declining trend during 2003–2017, likely due to residual degradation effects (see Fig. 9c and Fig. 10).

Secondly, our harmonization strategy uses GOME-2A as the reference sensor. Its extended data record (2007–2021) provides over five years of overlap with both SCIAMACHY and OCO-2, allowing us to perform a single-step normalization for each. This approach helps reduce the uncertainty propagation associated with multi-step corrections. In contrast, the LT_SIFc* product uses GOME as the benchmark, relying on only a six-month overlap with SCIAMACHY and then sequentially calibrating SCIAMACHY and GOME-2A, which may accumulate uncertainties.

To quantify the impact of overlap duration on harmonization uncertainty, we performed normalization experiments using 6-, 12-, and 24-month overlap periods between GOME-2 and OCO-2. Each experiment was repeated 10 times to assess the variability of the resulting harmonized time series.

[Figure]

Figure 2 Normalized SIF time series using different overlap durations: 6 months (first column), 12 months (second column), and 24 months (third column) from SCIAMACHY (top row) and OCO-2 (bottom row). The shaded areas represent the standard deviation (SD) across multiple experiments, with units of mW m$^{-2}$ sr$^{-1}$ nm$^{-1}$.

The results show that a six-month overlap leads to substantially higher variability (7.7% for SCIAMACHY and 8.0% for OCO-2) compared to longer overlaps. As the overlap period increased from 6 to 12 months, the standard deviations of the normalized SIF series decreased from 0.015 to 0.007 mW m$^{-2}$ sr$^{-1}$ nm$^{-1}$ (SCIAMACHY) and from 0.018 to 0.005 mW m$^{-2}$ sr$^{-1}$ nm$^{-1}$ (OCO-2), representing a reduction of over 53.3%. These results confirm that short overlap periods increase normalization uncertainty and highlight the robustness of our chosen strategy, which avoids using GOME directly as the baseline. Nevertheless, users should be aware of the potential uncertainties in the early portion (from 1995 to 2003) of the LHSIF record.

Furthermore, in contrast to the grid-specific matching methods adopted in previous studies, we implemented a region-based and month-specific normalization strategy. The region-based approach allows for a larger sample size within each region, potentially enabling a more robust estimation of the CDF, while the month-specific treatment helps account for seasonal variations in the CDF. The improved stability of interannual trends in the LHSIF product, compared to LT_SIFc* and SIF_005 at both global and regional scales (Fig. 10), may benefit from this normalization strategy.

Once again, we greatly appreciate the reviewer's helpful suggestions, which have led to substantial improvements in both the presentation and validation of our dataset.

**Major Comment 2:** Several aspects of the validation remain unclear

(1) Fig. 3: Why was only a single year (1998) selected for comparison to evaluate the data performance? This limited temporal scope may not sufficiently represent the overall dataset characteristics.

**Response:** Thanks for this comment. Although Fig. 3 presents results for a single year (1998), validation at other periods is also provided in the manuscript. For instance, Fig. 2 displays downscaling results for July 1996, and Fig. 4 shows the latitudinal distribution of validation errors across the full time series from 1995 to 2024. In the supplementary materials, Figs. S3–S5 present comparable results to Fig. 3 for the other three sensors in different years.

To further address the concern of insufficient data verification, we have included additional validation results. We have added histograms for the downscaling residuals (Fig. S2) in the revised version to provide a more quantitative view of the errors. These histograms are based on multiple years and multiple sensors, with the mean and standard deviation of residuals annotated in each panel to enhance the credibility and robustness of the validation. The newly added content is as follows:

**Main text:**

**3.1 Downscaled SIF dataset**

The comparison of fine-resolution (0.05°) and coarse-resolution (1°) SIF datasets, derived from GOME, is illustrated in Fig. 2. The top two rows (panels a–f) illustrate the enhanced spatial variability achieved through the downscaling process, revealing finer vegetation patterns and distinct intensity gradients. The downscaled SIF datasets render subtle patterns in SIF more apparent compared to the original coarse-resolution data (panels g–k). Additionally, the downscaling method, which incorporates neighborhood-based pixel searching, effectively fills in data gaps in the original data while preserving spatial continuity. The residual, which was calculated as the difference between the downscaled SIF (which was re-aggregated to original 1° resolution) and the original SIF is shown in panels l–p. It can be observed that in major vegetated regions, the residuals are concentrated within the range of -0.5 to 0.5 mW m$^{-2}$ sr$^{-1}$ nm$^{-1}$. The histograms of the downscaling residuals across different years and sensors are shown in Fig. S2. Overall, the absolute values of the mean residuals are less than 0.008 mW m$^{-2}$ sr$^{-1}$ nm$^{-1}$, and the standard deviations are below 0.105 mW m$^{-2}$ sr$^{-1}$ nm$^{-1}$.

https://doi.org/10.5194/essd-2025-94

[Figure]

Figure 3. Spatially downscaled SIF maps (a-f) compared to the original SIF maps at a coarser resolution (g–k). SIF data from GOME observations in July 1996 are shown as an example. **The bottom row (l–p) shows the downscaling residuals, which were calculated as the difference between original SIF and downscaled SIF.** Panels b, g, and l depict North America; c, h, and m focus on Europe; d, i, and n depict East Asia (centered on China); e, j, and o represent the Amazon Basin; and f, k, and p show Sub-Saharan Africa.

**Supplementary Materials:**

https://doi.org/10.5194/essd-2025-94

[Figure]

Figure S1 The Histograms of the downscaling residuals for GOME (1996, first row), SCIAMACHY (2008, second row), GOME-2 (2014, third row), and OCO-2 (2018, fourth row). The left, middle, and right columns correspond to results for January, July, and December, respectively.

These additions are intended to better illustrate the temporal stability and generalizability of the dataset across different time windows.

(2) Long-term trend validation: The analysis appears to rely solely on the global average SIF shown in Fig. 5. Furthermore, the description in Lines 275-280 is largely qualitative. More quantitative metrics (e.g., statistical significance tests or error metrics) would strengthen the validation.

**Response:** Thanks for this comment. In response, we have enriched Fig. 5 with additional quantitative information, including the mean and standard deviation of the global average SIF time series derived from each sensor as well as the harmonized product. The description in Lines 275–280 has also been revised to reflect these quantitative analyses. The updated figure and revised paragraph are provided below for your reference:

**3.2 Temporal harmonization**

https://doi.org/10.5194/essd-2025-94

The time series of the original SIF datasets from individual satellites and the resulting long-term harmonized SIF dataset (1995–2024) are presented in Fig. 5. Before normalization, substantial inter-sensor discrepancies were observed: mean SIF values ranged from 0.19 mW m⁻² sr⁻¹ nm⁻¹ (SCIAMACHY) to 0.28 mW m⁻² sr⁻¹ nm⁻¹ (GOME), while interannual trends varied from -0.76% yr⁻¹ (GOME) to 0.54% yr⁻¹ (GOME-2). Among the original sensor datasets, only the GOME-2 dataset showed a statistically significant trend ($p < 0.05$), whereas other sensors exhibited non-significant variations ($p \geq 0.05$). In contrast, the harmonized LHSIF dataset demonstrated a significant positive trend ($p < 0.001$, Trend = 0.31% yr⁻¹).

[Figure]

Figure 4. Global-averaged SIF time series derived from GOME (yellow), SCIAMACHY (blue), GOME-2 (green), and OCO-2 (red), along with the long-term harmonized SIF time series (LHSIF, gray dotted line), which aligns the satellite datasets based on overlapping periods. The table on the right summarizes the statistical characteristics of each sensor, including the mean, standard deviation (std), and the annual trend (Trend) averaged over the respective time spans. The statistical significance of the trends is indicated as follows: n.s. for not significant ($p > 0.05$), * for significant ($p < 0.05$), and *** for highly significant ($p < 0.001$).

(3) Fig. 8: What is the rationale for using annual maximum values rather than annual means? Maximum values are generally more susceptible to noise interference. Additionally, how was the claimed 'reduction in uncertainty' quantified?

**Response:** We appreciate the reviewer's insightful comment. We acknowledge the general concern that maximum values may be more sensitive to noise. We would like to clarify that in our study, the annual maximum SIF is defined as the composite value of the peak month (typically occurring during the June–August growing season) within each year, rather than an instantaneous measurement. This monthly compositing approach minimizes the influence of transient noise or outliers, thereby addressing the reviewer's concern.

We chose to use annual maximum SIF values over annual means for several reasons:

(1) The annual peak SIF better reflects vegetation productivity and ecosystem functioning, as it coincides with the peak growing season when photosynthetic activity is most active and data quality is highest.

(2) Satellite SIF products often suffer from low signal-to-noise ratios, especially in sparsely vegetated regions or during dormant seasons. Using mean values over the full year may be disproportionately affected by these low-quality periods.

Based on these considerations, the use of annual maximum SIF was retained in the revised manuscript.

https://doi.org/10.5194/essd-2025-94

Regarding the "reduction in uncertainty" mentioned in the original manuscript, we refer to the decrease in the standard deviation of the fitted trend after normalization. This has been clarified in the revised manuscript as below:

The annual maximums of the global-averaged SIF were used to investigate the fluctuation of the worldwide vegetation from 1995 to 2024. Significant interannual fluctuations were found for the SIF time series without normalization, with an overall decline (blue line in Fig. 7a). **The normalized SIF time series reveals a growth rate of 0.31% yr$^{-1}$. After normalization, the standard deviation of the fitted slope decreases from 0.25% to 0.07%, indicating a reduction in uncertainty.** The boxplot in Fig. 7b further shows a narrower range of SIF values after temporal normalization, suggesting a more concentrated data distribution and improved comparability across sensors.

[Figure]

SIF$_u$: Trend = -0.52 ± 0.25 % yr$^{-1}$

SIF$_c$: Trend = 0.31 ± 0.07 % yr$^{-1}$

Figure 5. (a) Trend and (b) box plot of the yearly maximum global-averaged SIF of the combined time series before (SIF$_u$) and after (SIF$_c$) normalization during 1995–2024

(4) Temporal inconsistency: Why does Fig. 8 cover 1996 – 2023 while Fig. 9 starts from 1995? This discrepancy in time ranges should be explicitly justified.

**Response:** Thanks for pointing out the inconsistency. The difference in time ranges arises from the use of different statistical metrics in Figs. 8 and 9. Specifically, Fig. 8 shows annual averages, while Fig. 9 depicts the trend of annual peak-month values (usually observed in July). Since satellite observations in 1995 only began in July, that year was excluded from Fig. 8 to avoid bias due to incomplete coverage. In contrast, the available data in 1995 were sufficient to determine the peak for that year, allowing its inclusion in the analysis in Fig. 9.

To clarify this point, we have added the following explanation to the caption of Fig. 8 in the revised manuscript:

[Figure]

Figure 6. (a) Map of trends in LHSIF for 1996–2024. (b) Percentage of areas in global vegetation covered by four different trend types (significant decrease: negative correlation and $p < 0.05$; decrease: negative correlation and $p \geq 0.05$; increase: positive correlation and $p \geq 0.05$; and significant increase: positive correlation and $p < 0.05$). The black dots in (a) represent statistically significant trends ($p < 0.05$). **The statistics begin in 1996 due to incomplete data coverage in 1995, which only includes the second half of the year (July–December).**

To clarify, these observations are not meant to imply that the authors' approaches are inherently flawed. However, clearer explanations for these methodological choices are necessary to ensure robust interpretation of the results.

**Response:** We thank the reviewer for the clarification. We agree that clearer explanations can strengthen the interpretation of the results. Accordingly, we have revised the manuscript to provide more quantitative support for the dataset developed in this study.

**Major Comment 3:** In Section 4.1, the authors show that the overlap between GOME and SCIAMACHY SIF records is limited to only six months. However, the manuscript lacks a detailed explanation for addressing this issue within the current study. In fact, the challenge of short-term overlap is also present in this study. The limited temporal overlap constrains the representativeness of the GOME SIF data and may introduce uncertainty and potential biases in the mean-standard deviation-based matching approach. This issue is evident in Figures 7 and 8. For instance, in Figure 7, noticeable discrepancies between the corrected and uncorrected datasets are observed prior to 2003. It is worth noting that these early-period differences reflect the extent to which the correction improved the quality of this dataset. Therefore, I suggest that the authors further clarify the applicability and effectiveness of this matching method in mitigating biases arising from short-term overlaps, and possibly, supplement the analysis with a quantitative assessment of associated uncertainties.

**Response:** Thank you for this valuable comment. We agree that the limited temporal overlap between sensors can introduce potential uncertainties in the harmonization process. In the revised manuscript, we have addressed this issue by explicitly testing how the length of the overlap period affects the performance of the CDF-based matching method. Specifically, we conducted a set of quantitative experiments to evaluate the uncertainty associated with short overlap durations. The relevant additions can be found in Section 4.1. The results underline the potential uncertainty associated with the early part of the SIF time series (1995–2003).

https://doi.org/10.5194/essd-2025-94

We have added a cautionary note to alert readers to the greater uncertainty in this early period due to the short overlap between GOME and SCIAMACHY, summarized as follows:

**4.1 Improvements in cross-sensor harmonization**
* * *
Secondly, our harmonization strategy uses GOME-2A as the reference sensor. Its extended data record (2007–2021) provides over five years of overlap with both SCIAMACHY and OCO-2, allowing us to perform a single-step normalization for each. This approach helps reduce the uncertainty propagation associated with multi-step corrections. In contrast, the LT_SIFc* product uses GOME as the benchmark, relying on only a six-month overlap with SCIAMACHY and then sequentially calibrating SCIAMACHY and GOME-2A, which may accumulate uncertainties.

To quantify the impact of overlap duration on harmonization uncertainty, we performed normalization experiments using 6-, 12-, and 24-month overlap periods between GOME-2 and SCIAMACHY/OCO-2 (Fig. 13). Each experiment was repeated 10 times to assess the variability of the resulting harmonized time series.

[Figure]

Figure 7 Normalized SIF time series using different overlap durations: 6 months (first column), 12 months (second column), and 24 months (third column) from SCIAMACHY (top row) and OCO-2 (bottom row). The shaded areas represent the standard deviation (SD) across multiple experiments, with units of mW m$^{-2}$ sr$^{-1}$ nm$^{-1}$.

The results show that a six-month overlap leads to a higher standard deviation in SIF time series compared to longer overlaps. As the overlap period was extended from 6 to 12 months, the standard deviations of the normalized SIF series decreased from 0.015 to 0.007 mW m$^{-2}$ sr$^{-1}$ nm$^{-1}$ (SCIAMACHY) and from 0.018 to

https://doi.org/10.5194/essd-2025-94

0.005 mW m$^{-2}$ sr$^{-1}$ nm$^{-1}$ (OCO-2), representing a reduction of over 53.3%. **These results confirm that short overlap periods increase normalization uncertainty and highlight the robustness of our chosen strategy, which avoids using GOME as the baseline. Nevertheless, users should be aware of the potential uncertainties in the early portion (from 1995 to 2003) of the LHSIF record.**

**Major Comment 4:** The authors' descriptions in multiple sections of the manuscript are overly qualitative and subjective, lacking sufficient experimental support. It is recommended that the authors revise the relevant language to provide more detailed explanations or supplement with additional experiments.

Line 86; 268-269; 391; 394; 398;

**Response:** We appreciate the reviewer's suggestion. We have revised the manuscript to reduce subjective or qualitative expressions and incorporated additional quantitative analyses to support our conclusions.

First, we introduced quantitative indicators for time series trend analysis in the methods section, as detailed below:

**3.4 Temporal Trend Analysis Metrics**

To assess long-term trends in vegetation dynamics, we employed the Mann-Kendall (MK) test, a non-parametric method suitable for detecting monotonic trends in time series data, using the Python package pyMannKendall (Hussain and Mahmud, 2019). Trend estimation uncertainty was quantified by the standard deviation and 95% confidence intervals of the estimated temporal trend.

The detailed modifications are as follows:

**(1) Line 86**

**Original:**

"A precise, reliable, harmonized, and global high-resolution SIF dataset is not yet available for long-term vegetation monitoring."

**Revision:**

To avoid emphasizing subjective terms such as "precise," "reliable," and "harmonized" in the introduction, we removed this sentence. Instead, we revised the text as follows:

So far, the cross-sensor consistency of existing long-term SIF records remains to be further evaluated. In this study, we employed the TCSIF dataset as a physically calibrated benchmark to constrain the long-term consistency of GOME, SCIAMACHY, and OCO-2 SIF observations.

We also added more quantitative accuracy assessments and comparisons with other products in Section 3.3 of the revised manuscript to support the reliability of the LHSIF dataset (please refer to the response to Major comment 1).

**(2) Line 268-269**

**Original:**

"As shown by the temporally averaged residuals (Fig. 4 (b)), the absolute mean residuals for most locations are less than 0.05 mW m$^{-2}$ sr$^{-1}$ nm$^{-1}$ for regions below 70° N. This result highlights the global applicability of the LUE-based downscaling method."

**Revision:**

We added quantitative descriptions and replaced the subjective statement with a more informative explanation:

The temporal and spatial distributions of the spatial downscaling residuals were analyzed (Fig. 4). **The monthly mean residuals across different latitudes and months were generally below 0.2 mW m$^{-2}$ sr$^{-1}$ nm$^{-1}$ (Fig. 4 a). In addition, the regions with relatively larger residuals (e.g., > 0.1 mW m$^{-2}$ sr$^{-1}$ nm$^{-1}$) were**

**mainly located in high-latitude areas.** As shown by the temporally averaged residuals (Fig. 4 b), for most areas below 70°N, the absolute mean residuals are less than 0.05 mW m$^{-2}$ sr$^{-1}$ nm$^{-1}$ for regions below 70° N. **These results indicate that the downscaling method maintains high consistency with the original data across a broad range of temporal and spatial conditions.**

**(3) Line 391-394**

**Original:**

"The approach proposed adopted in this study is robust compared to CDF matching. By modifying only the mean and standard deviation of the SIF data distribution, the method avoids overfitting and enhances stability. This less stringent approach ensures harmonization while preserving the inherent variability of the datasets. Therefore, the LHSIF dataset provides an unprecedented long-term harmonized SIF dataset that is theoretically reliable and demonstrates a reasonable temporal trend."

**Revision:**

We have revised the manuscript to remove the overstatement about the reliability of the method. Rather than emphasizing the superiority of our approach, we present additional analytical results to objectively demonstrate its advantages. In fact, the revised version adopts a CDF-based harmonization method similar to that used by Wang et al., and therefore, we have removed the original subjective claims regarding methodological robustness.

Instead, we have added a comparative analysis of the seasonal variations derived from LHSIF and LT_SIFc* products to highlight the improved consistency and continuity of our results. The relevant revisions are detailed below:

The interannual trends of several long-term SIF products—including LHSIF, LT_SIFc*, SIF_005, and LCSIF—were compared. The annual maximum of global monthly SIF was used for comparison. Fig. 10 presents the results for the global scale as well as for several representative climate zones and land cover types.

Among the four SIF products, all except SIF_005 show increasing global trends. LHSIF exhibits the strongest upward trend at 0.31% yr$^{-1}$, while LCSIF presents the most stable interannual variation, with a trend standard deviation of only 0.01% yr$^{-1}$. LHSIF and LCSIF display statistically significant increases on the global scale, whereas the trends for LT_SIFc* and SIF_005 are not statistically significant. The divergent trend between SIF_005 and the other SIF products is further demonstrated on regional scales. For example, in continental cropland regions (Fig. 10d) and temperate deciduous broadleaf forest (DBF) biomes, LHSIF, LT_SIFc*, and LCSIF generally exhibit consistent positive trends, whereas SIF_005 shows a declining trend.

In most cases shown in Fig. 10, LHSIF, LT_SIFc*, and LCSIF display consistent trends. An exception occurs in arid regions, where LCSIF shows an increasing trend while both LHSIF and LT_SIFc* exhibit decreasing trends (Fig. 10j,k). This divergence may be attributed to the machine learning–based nature of LCSIF, which relies heavily on predictor variables and may not fully capture the actual SIF dynamics under stress conditions. In contrast, the observational basis of LHSIF enables it to more directly reflect regional responses to environmental variability.

[Figure]

Figure 8. Comparison of interannual variations in long-term SIF products. LHSIF (red), LT_SIFc* (green), SIF_005 (purple), and LCSIF (blue) are compared for (a) the global scale and (b–m) various climatic and vegetation regions. All datasets were normalized using the z-score method. Dashed lines represent yearly maximum values, and solid lines indicate linear trends. To aid visual comparison, trend lines were anchored at the origin (2010, 0). The statistical significance of the trends is indicated as follows: n.s. for not significant (p > 0.05), * for significant (p < 0.05), and *** for highly significant (p < 0.001).

**(4) Line 398**
**Original:**

Despite its high performance, it is crucial to remember that this statistical methodology is not a rigorous calibration procedure.
**Revision:**

We added a more detailed explanation, as shown below:

Although the normalized series yields consistent seasonal and interannual trends across sensors, it is important to note that this normalization approach does not involve a physically based calibration (e.g., radiometric calibration based on pseudo-invariant targets). As a result, residual sensor-specific biases may still exist.

https://doi.org/10.5194/essd-2025-94

**Major Comment 5:** To more comprehensively evaluate the quality of the LHSIF dataset, it is recommended that the authors incorporate the LCSIF dataset (Fang et al., 2025) for comparative experiments in their study. Comparing these two datasets can reveal the potential strengths and limitations of LHSIF, particularly in terms of long-term trend analysis and responses across different ecosystems. Furthermore, the long-term record of LCSIF can serve as a reference to validate the consistency and accuracy of LHSIF data during the earlier years.

**Response:** Thanks for this valuable suggestion. In response, we have incorporated a comparative analysis between the LHSIF and LCSIF datasets, including assessments of long-term trends and ecosystem-specific responses. The corresponding results have been added to the revised manuscript as below:

The interannual trends of several long-term SIF products—including LHSIF, LT_SIFc*, SIF_005, and LCSIF—were compared. The annual maximum of global monthly SIF was used for comparison. Fig. 10 presents the results for the global scale as well as for several representative climate zones and land cover types.

Among the four SIF products, all except SIF_005 exhibit increasing global trends. LHSIF shows the strongest upward trend at 0.31% yr$^{-1}$, **while LCSIF presents the most stable interannual variation, with a trend standard deviation of only 0.01% yr$^{-1}$. LHSIF and LCSIF display statistically significant increases on the global scale, whereas the trends for LT_SIFc* and SIF_005 are not statistically significant.** The divergent trend between SIF_005 and the other SIF products is further demonstrated on regional scales. For example, in continental cropland regions (Fig. 10d) and temperate deciduous broadleaf forest (DBF) biomes, LHSIF, LT_SIFc*, and LCSIF generally exhibit consistent positive trends, whereas SIF_005 shows a declining trend.

**In most cases shown in Fig. 10, LHSIF, LT_SIFc*, and LCSIF exhibit consistent trends. An exception occurs in arid regions, where LCSIF shows an increasing trend while both LHSIF and LT_SIFc* exhibit decreasing trends (Fig. 10j,k). This divergence may be attributed to the machine learning–based nature of LCSIF, which relies heavily on predictor variables and may not fully capture the actual SIF dynamics under stress conditions. In contrast, the observational basis of LHSIF enables it to more directly reflect regional responses to environmental variability.**

https://doi.org/10.5194/essd-2025-94

[Figure]

Figure 9. Comparison of interannual variations in long-term SIF products. LHSIF (red), LT_SIFc* (green), SIF_005 (purple), and LCSIF (blue) are compared for (a) the global scale and (b–m) various climatic and vegetation regions. All datasets were normalized using the z-score method. Dashed lines represent yearly maximum values, and solid lines indicate linear trends. To aid visual comparison, trend lines were anchored at the origin (2010, 0). The statistical significance of the trends is indicated as follows: n.s. for not significant ($p > 0.05$), * for significant ($p < 0.05$), and *** for highly significant ($p < 0.001$).

It is important to note that the LHSIF product is derived solely from observed satellite SIF measurements, aiming to preserve the original signal characteristics to the greatest extent possible. In contrast, machine learning-based SIF products, such as LCSIF, are generated through data-driven modeling approaches and represent simulations of SIF signals, rather than direct satellite observations. While machine learning-based reconstructions often yield smoother time series, this smoothness does not necessarily imply higher fidelity. In some cases, it may result from overfitting seasonal patterns, potentially suppressing high-frequency variations that carry meaningful biophysical signals(Ma et al., 2022; Chen et al., 2025).

**Detailed Comment 1:** Line 17: How did you define the reduction in overall error? Compared to what?
**Response:** Thanks for this comment. We have added the following supplementary explanation in the abstract.

The resulting harmonized dataset shows a 49% reduction in inter-sensor differences compared to the uncorrected data and exhibits a stable interannual increase of $0.31 \pm 0.07\%$ yr$^{-1}$.

A detailed explanation of the result can be found in Sec. 3.2 as follows:

In all tested scenarios, the normalization process substantially reduced the differences between the two sensors. Overall, the MSD decreased by more than 49% following normalization.

**Detailed Comment 2:** Line 27: garnered significant attention(s)

**Response:** Thanks, it has been corrected.

**Detailed Comment 3:** The second and third paragraphs can be merged to one paragraph after reducing some sentences about SIF-GPP relationship which is not the focus in this study.

**Response:** Thanks for this comment. These two paragraphs have been revised as below:

Following the publication of the initial global SIF map from the Greenhouse Gases Observing Satellite (GOSAT), interest in the SIF-GPP association greatly increased (Frankenberg et al., 2011; Guanter et al., 2012; Joiner et al., 2011). Subsequent satellite-based analyses have consistently revealed strong spatial and temporal correlations between SIF and GPP, showcasing remarkable alignment between SIF and GPP in terms of spatial distribution and seasonal variability (Anav et al., 2015; Li et al., 2018; Verma et al., 2017; Yang et al., 2015; Guanter et al., 2014; Zheng et al., 2024). However, these results are mostly based on coarse-resolution SIF datasets such as the Global Ozone Monitoring Experiment (GOME)-2, leading to potential spatial mismatch issues. Additionally, the SIF-GPP link varies by vegetation type, emphasizing the critical need for SIF datasets with higher spatial resolution and spatiotemporal consistency to better support ecosystem monitoring and interpretation.

**Detailed Comment 4:** Line 65: delete "itself"

**Response:** Thanks, it has been deleted.

**Detailed Comment 5:** Line 70-74: Could you provide a direct explanation of where Wang's research may have fallen short or failed to consider comprehensively?

**Response:** Thanks for this comment. We strengthened the limitation of Wang's research in the revised manuscript as below:

To mitigate this limitation, Wang et al. (2022) attempted to create a temporally corrected long-term SIF product (LT_SIFc*) by correcting the degradation trends in gridded GOME, SCIAMACHY, and GOME-2 SIF products. **However, the method lacks a physically based correction of the actual sensor radiance degradation and instead applies adjustments at the SIF product level, which may not accurately reflect the true instrumental change. This is further complicated by** the nonlinear characteristics inherent in the retrieval methodology and subsequent processing procedures (e.g., zero-bias correction and quality filtering), which prevent a direct and linear propagation of sensor degradation into the final SIF values. Recently, the temporally corrected GOME-2A SIF dataset (TCSIF) included a calibration of the radiance measurements of GOME-2A using a pseudo-invariant method (Zou et al., 2024). This correction effectively eliminates the influence of sensor degradation over time, providing a robust benchmark for generating long-term harmonized SIF products.

**Detailed Comment 6:** Line 84-85: Wen's research has significant overlap with yours, so I'm unclear why it was only mentioned so briefly and at such a late stage.

**Response:** Thanks for this comment. We acknowledge the importance of Wen's research and have revised the introduction as below:

Despite the availability of multiple satellite SIF products, most have a temporal coverage shorter than 10 years, and large discrepancies have been observed between different SIF products (Parazoo et al., 2019). These

https://doi.org/10.5194/essd-2025-94

temporal inconsistencies may stem from differences in retrieval algorithms, absolute radiometric calibration errors, instrumental artifacts, directional effects, and variations in satellite overpass times and footprint sizes (Zhang et al., 2018b; Bacour et al., 2019). **To address these challenges, Wen et al. (2020) proposed a harmonization framework that used cumulative distribution function (CDF) to integrate SIF datasets from SCIAMACHY and GOME-2 during their overlapping period, resulting in a continuous record from 2002 to 2018.**

**While Wen's framework laid the foundation for cross-sensor harmonization, it primarily focused on aligning overlapping periods and did not explicitly address instrument degradation, a key factor that compromises the long-term consistency of single-sensor records.** Such degradation, as observed in GOME-2, poses a significant challenge for long-term consistency and introduces uncertainties in trend analyses (Parazoo et al., 2019). For instance, Yang et al. (2018) reported diverging trends between EVI and SIF, attributing the latter's decline to reduced photosynthetic activity. However, Zhang et al. (2018a) argued that this conclusion was impacted by the deterioration of the GOME-2A instrument. Further research by Koren et al. (2018) showed that the decline in SIF persisted even after correcting for sensor degradation. The SIFTERv2 product (Van Schaik et al., 2020) employed in Koren's study was simply corrected using linear models; the reliability of SIFTERv2 decreased significantly after 2016, limiting its application for long-term trend analysis.

To mitigate this limitation, Wang et al. (2022) attempted to create a temporally corrected long-term SIF product (LT_SIFc*) by correcting the degradation trends in gridded GOME, SCIAMACHY, and GOME-2 SIF products. However, the nonlinear characteristics inherent in the retrieval methodology and subsequent processing procedures (e.g., zero-bias correction and quality filtering) presented a challenge. The bias in sensor observations was not linearly transferred to the SIF product. This approach risks introducing inaccuracies, because the trend corrected in the SIF product may not correspond to the true sensor degradation. Recently, the temporally corrected GOME-2A SIF dataset (TCSIF) included a calibration of the radiance measurements of GOME-2A using a pseudo-invariant method (Zou et al., 2024). This correction effectively eliminates the influence of sensor degradation over time, providing a robust benchmark for generating long-term harmonized SIF products.

**Detailed Comment 7:** Line 87: 'A precise, reliable, harmonized, and global high-resolution SIF dataset is not yet available for long-term vegetation monitoring.' Based on the preceding sections of the introduction, the authors do not appear to have clearly explained why the prior data were neither precise nor reliable.

**Response:** Thanks for this comment. We have revised the introduction to more clearly articulate the limitations of previous datasets, thereby clarifying the motivation for developing the LHSIF product. The revised text is as follows:

**While Wen's framework laid the foundation for cross-sensor harmonization, it did not explicitly address instrument degradation—a key factor that compromises the long-term consistency of single-sensor records. Such degradation, as observed in GOME-2, poses a significant challenge for long-term consistency and introduces uncertainties in trend analyses (Parazoo et al., 2019).** For instance, Yang et al. (2018) reported diverging trends between EVI and SIF, attributing the latter's decline to reduced photosynthetic activity. However, Zhang et al. (2018a) argued that this conclusion was impacted by the deterioration of the GOME-2A instrument. Further research by Koren et al. (2018) showed that the decline in SIF persisted even after correcting for sensor degradation. The SIFTERv2 product (Van Schaik et al., 2020) employed in Koren's study was simply corrected using linear models; the reliability of SIFTERv2 decreased significantly after 2016, limiting its application for long-term trend analysis.

To mitigate this limitation, Wang et al. (2022) attempted to create a temporally corrected long-term SIF product (LT_SIFc*) by correcting the degradation trends in gridded GOME, SCIAMACHY, and GOME-2

https://doi.org/10.5194/essd-2025-94

SIF products. **However, the method lacks a physically based correction of the actual sensor radiance degradation and instead applies adjustments at the SIF product level, which may not accurately reflect the true instrumental change.** This is further complicated by **the nonlinear characteristics inherent in the retrieval methodology and subsequent processing procedures (e.g., zero-bias correction and quality filtering), which prevent a direct and linear propagation of sensor degradation into the final SIF values.** Recently, the temporally corrected GOME-2A SIF dataset (TCSIF) included a calibration of the radiance measurements of GOME-2A using a pseudo-invariant method (Zou et al., 2024). This correction effectively eliminates the influence of sensor degradation over time, providing a robust benchmark for generating long-term harmonized SIF products.

**So far, the cross-sensor consistency of existing long-term SIF records remains to be further evaluated.**

**Detailed Comment 8:** Line 131: Oco-2/Oco-3 → OCO-2/OCO-3

**Response:** Thanks, it has been corrected.

**Detailed Comment 9:** Line 207: Is this spatial resolution (0.072727° × 0.072727°) commonly referred to?

**Response:** Thanks for this comment. Yes, this spatial resolution is commonly used in related studies. Please refer to (Chen et al., 2019)and (He et al., 2021) for more details.

**Detailed Comment 10:** Line 221: There appear to be numerous NDVI products based on AVHRR data. Is this a newly developed set? Could you please provide some additional introduction?

**Response:** Thanks for this comment. We have added more information in Sec. 2.5.3 as below:

**2.5.3 AVHRR vegetation indices**

Global NDVI and NIRv datasets from 1995 to 2021, derived from the AVHRR sensors, were utilized in this study. These datasets were developed by Jeong et al. (2024) based on the AVHRR Long-Term Data Record version 5 (LTDR V5) surface reflectance product. **To address temporal inconsistency in long-term AVHRR records, a three-step correction was applied, including cross-sensor calibration, orbital drift correction, and machine learning-based harmonization with MODIS vegetation indices. This post-processing significantly improved the temporal consistency of NDVI and NIRv from 1982 to 2021, as verified using detrended anomalies and trends at calibration sites. The final product enables more robust analyses of long-term vegetation dynamics and reduces spurious trends due to sensor artifacts.**

**Detailed Comment 11:** Line 238-243: Could be more **quantitative**

**Response:** Thanks for this comment. To improve the quantitative evaluation of the downscaling results, we have revised Figure 3 and added additional statistical indicators, including RMSE and the fitted regression line. The updated figure and its description are as follows:

The distribution of monthly SIF before and after spatial downscaling is shown using GOME as an example (Fig. 3), while results for the other three satellites are provided in Figs. S3–S5. The spatially downscaled SIF (0.05° × 0.05°) was re-aggregated to 1° × 1° or 0.5° × 0.5° resolution for comparison with the original coarse-resolution SIF. **The results demonstrate that the SIF values from the re-aggregated pixels are generally consistent with the original SIF values, closely clustering along the 1:1 line and showing strong agreement ($R^2 > 0.73$, RMSE $< 0.11$ mW m$^{-2}$ sr$^{-1}$ nm$^{-1}$),** indicating that the LUE-based downscaling model effectively captures the relationship between SIF and its driving variables.

[Figure]

Figure 10. The relationship between the reaggregated GOME SIF (SIF_reagg) and the original GOME SIF (SIF_original) for 1998 (by month).

In addition, we sincerely apologize that the previous version of Figure 3 was generated using an outdated version of the GOME dataset due to a data management oversight. This has now been corrected: the figure has been re-generated using the updated and consistent dataset. Importantly, this correction does not affect the main conclusions of the study, but it improves the accuracy and consistency of the results.

**Detailed Comment 12:.** Figure 5: the order of GOME and SCIAMACHY in the legend is opposite.
**Response:** Thanks for this comment. We have checked and confirmed that the legend labels are now correct in the revised version.

**Detailed Comment 13:.** Line 267: as the difference between ? some words missed
**Response:** Thanks for this comment. The missing explanation has been added as follows:
The residuals are calculated as the difference between the reaggregated SIF (SIF_reagg) and the original SIF (SIF_original).

**Detailed Comment 14:** Line 340-345: These belong to Methods. Please check throughout the manuscript.
**Response:** Thanks for this comment. Corresponding sentences have been moved to Sec. 2.5.4 as below:

In addition, tower-based SIF observations from the ChinaSpec network, including sites such as DM, GC, HL, XTS, and AR (Zhang et al., 2021), were used to validate the accuracy and spatiotemporal consistency of

the long-term SIF dataset generated in this study. The locations and cover types of the ChinaSpec sites used are listed in Table S1. **To ensure consistent comparisons, the tower-based SIF at 760.6 nm was converted to 740 nm using an empirical correction factor of 1.48 (Du et al., 2023). Additionally, the original half-hourly tower-based SIF data were temporally upscaled to daily and monthly values with the aid of PAR and NDVI, following the method described by Hu et al. (2018).**

**Detailed Comment 15:** Line 343: please add the citation for this factor

**Response:** Thanks for this comment. We have added the citation below:

To ensure consistent comparisons, the tower-based SIF at 760.6 nm was converted to 740 nm using an empirical correction factor of 1.48 **(Du et al. 2024).**

**Detailed Comment 16:** Lines 26, 144, 158, etc.: please add the space between the text and the brackets

**Response:** Thanks for this comment. Spaces have been inserted before the brackets throughout the manuscript.

**Detailed Comment 17:** Section 4.1: Could you provide more concrete examples demonstrating the advantages of using TCSIF compared to the uncorrected GOME-2A data employed in the other two studies?

**Response:** Thanks for this comment. As demonstrated in Zou et al. (2024), the TCSIF product shows a more stable interannual trend during the GOME-2A period (2007–2021): $0.705 \pm 0.152\%$ $yr^{-1}$, compared to $1.247 \pm 0.234\%$ $yr^{-1}$ in LT_SIFc* and $-0.078 \pm 0.356\%$ $yr^{-1}$ in SIF_005. The advantages of TCSIF are also observed in the LHSIF dataset, which displays more consistent trends (see Fig. 10). In contrast, SIF_005 shows a declining trend in several regions, likely due to uncorrected sensor degradation.

As the temporal correction is not the focus of this study, we respectfully refer the reviewer to Zou et al. (2024) for a more detailed comparison and methodology.

**Detailed Comment 18:** Line 374: Missing word.

**Response:** Thank you for pointing this out. The sentence at Line 374 was removed during revisions made in response to other suggestions.

**Detailed Comment 19:** Line 385-387: Providing concrete examples rather than textual descriptions would enable readers to better understand.

**Detailed Comment 20:** Line 391: Please see above.

**Response:** Thanks for these comments. To address these concerns, we revised the relevant sentences to avoid subjective descriptions and instead provided clearer quantitative comparisons. Additional supporting analyses have been incorporated into Section 3.3 and the discussion part. For more details, please refer to our responses to Major Comment 1 and Major Comment 5.

The references cited in the response are listed below:

Chen, J. M., Ju, W., Ciais, P., Viovy, N., Liu, R., Liu, Y., and Lu, X.: Vegetation structural change since 1981 significantly enhanced the terrestrial carbon sink, Nature Communications, 10, 4259, 10.1038/s41467-019-12257-8, 2019.

Chen, S., Liu, L., Sui, L., Liu, X., and Ma, Y.: An improved spatially downscaled solar-induced chlorophyll fluorescence dataset from the TROPOMI product, Scientific Data, 12, 135, 2025.

https://doi.org/10.5194/essd-2025-94

Didan, K.: MODIS/Terra Vegetation Indices 16-Day L3 Global 1km SIN Grid V006. [dataset], https://doi.org/10.5067/MODIS/MOD13A2.006, 2015.

He, Q., Ju, W., Dai, S., He, W., Song, L., Wang, S., Li, X., and Mao, G.: Drought Risk of Global Terrestrial Gross Primary Productivity Over the Last 40 Years Detected by a Remote Sensing-Driven Process Model, Journal of Geophysical Research: Biogeosciences, 126, e2020JG005944, https://doi.org/10.1029/2020JG005944, 2021.

Hu, J., Liu, L., Guo, J., Du, S., and Liu, X.: Upscaling Solar-Induced Chlorophyll Fluorescence from an Instantaneous to Daily Scale Gives an Improved Estimation of the Gross Primary Productivity, Remote Sensing, 10, 1663, 2018.

Ma, Y., Liu, L., Liu, X., and Chen, J.: An improved downscaled sun-induced chlorophyll fluorescence (DSIF) product of GOME-2 dataset, European journal of remote sensing, 55, 168-180, 2022.

Zou, C., Du, S., Liu, X., and Liu, L.: TCSIF: a temporally consistent global Global Ozone Monitoring Experiment-2A (GOME-2A) solar-induced chlorophyll fluorescence dataset with the correction of sensor degradation, Earth Syst. Sci. Data, 16, 2789-2809, 10.5194/essd-16-2789-2024, 2024.

---

## Author Comment (AC3)

https://doi.org/10.5194/essd-2025-94
**Response to Reviewer 1 Comments:**

This paper produced a long-term SIF dataset (LHSIF) from 1995 to 2023. This topic is of significant importance for global-scale carbon simulation and vegetation studies. I think the datasets preprocessing procedures are solid. However, there are some questions I am concerned, especially about the reliability of the inter-annual trend in the SIF data from 1995 to 2000.

**Response:** We sincerely thank the reviewer for recognizing the significance of our work and the robustness of the preprocessing procedures. We fully understand the reviewer's concern regarding the reliability of the inter-annual trend in the early part of the time series (1995–2000). In response, we have added a dedicated discussion in the manuscript to clarify the limitations and uncertainties associated with this period. Please see below for detailed responses. Reviewer comments are shown in black, authors' responses in blue, and the corresponding revisions to the manuscript in purple.

There are many spelling errors in sentences throughout the paper. I listed as much as I could found. Please double check to avoid such errors and also try to avoid using 'Figure.... illustrated...' as the begining of a paragraph in the result section. Following a logical writing orders is necessary.

I recommend minor revision and here are some comments I wish the authors could give explanations before accepting this paper for publication.

**Response:** We sincerely thank the reviewer for the detailed and constructive comments. We have carefully reviewed the entire manuscript to correct the identified language issues and have revised or rewritten problematic expressions accordingly. Detailed responses to each comment are provided below.

Specific concerns and suggestions are outlined as follows:

**Comment 1:** You didn't mention how do you do quality control for GPP provides by FLUXNET. Did you use the quality flag data of "GPP_DT_VUT_REF" during SIF production?

**Response:** Thank you for this comment. In the previous version of our manuscript, we selected GPP data that were neither missing nor equal to -999 for comparison. In the revised version, we have incorporated an additional quality control step by using the "NEE_VUT_REF_QC" field as a filter. Specifically, only GPP data points with QC values greater than 0.7 were retained to ensure the reliability of the data used for SIF product validation. Corresponding descriptions have been added to Section 2.5.4, as follows:

**2.5.4 Ground-based observations**

Ground-based SIF and GPP observations were integrated into this study to validate and enhance the interpretation of satellite-derived datasets. Specifically, FLUXNET GPP observations were employed, which are based on in-situ measurements from a global network of flux towers distributed across diverse ecosystems (Pastorello et al., 2020). FLUXNET sites with more than five years of data were grouped into climate zones and vegetation functional types (see Fig. S1 for site distribution and types). The field "GPP_DT_VUT_REF" was used. **To ensure the quality of the GPP data used for validation, only GPP records with the quality flag greater than 0.7 (Verma et al., 2015) were retained in this study.**

The correlation between SIF and GPP was further improved after applying quality filtering. The corresponding figure has been updated accordingly (originally Figure 10, now revised as Figure 11).

**3.3 Validation and comparative analysis**
* * *
In addition to long-term satellite products, ground-based observations were also incorporated for comparison.

https://doi.org/10.5194/essd-2025-94

The relationships of LHSIF and AVHRR NDVI with FLUXNET GPP are illustrated in Figure . The LHSIF product shows a strong ability to track GPP, especially for cropland and mixed forest types (Fig. 11a). In contrast, NDVI consistently exhibits lower $R^2$ values (Fig. 11b) and a more pronounced nonlinear relationship with GPP due to saturation effects. Apart from a few groups in the Southern Hemisphere (such as grasslands in tropical and arid areas), where only a small number of sites are available (see Fig. S1), SIF outperforms NDVI in most cases.

[Figure]

**Figure 11. Comparison of long-term relationships between SIF vs. GPP and NDVI vs. GPP at FLUXNET sites. Only those flux tower sites that have accumulated over a decade of data were chosen and subsequently categorized according to their respective climate zones and vegetation types.**

**Comment 2:** I am concerned whether the interannual trend in SIF before 2000 is true or reliable. For example, in Figure 7, we did not see much difference in yearly maximum global-averaged SIF based on the SIFu or SIFc from 2000 to 2023. The large discrepency mainly occurs in year 1995 to 2000, when SIFu is significant higher than the SIFc with a significnatly decreasing trend (mainly provided by GOME?). However, there exists larege discrenpency between SIFu and SIFc before 2000. How could you convince the readers that such inter-annual trend in SIF in year 1995 to 2000 is reliable? There is no validation of this trend from 1995 to 2000 as far as I can see in the paper.

**Response:** Thanks for this thoughtful comment. We appreciate the concern regarding the reliability of the interannual SIF trend prior to 2000. First, we would like to clarify that our normalization approach adjusts only the overall scale of the datasets based on their statistical relationships during the overlap period. It does not impose artificial changes on the original temporal trends.

To further investigate whether the decreasing trend during the early GOME period (1995–2000) is attributable to sensor degradation, we examined long-term reflectance at pseudo-invariant calibration sites (PICs) located in the Sahara and Arabian deserts (Fig. R1).

https://doi.org/10.5194/essd-2025-94

[Figure]

Figure R1 Locations of the PICs used to evaluate GOME sensor stability.

[Figure]

Figure R2 Long-term trends of GOME 780 nm reflectance and NDVI over the PICs.

As shown in Fig. R2, no significant decline was found in the GOME NIR-band (780 nm) reflectance or the NDVI. These results suggest that the GOME sensor did not exhibit obvious degradation in reflectance at relevant wavelengths. Therefore, we did not apply an artificial trend correction to the GOME SIF record.

However, since GOME and SCIAMACHY share only a six-month overlap period, such a short duration may introduce uncertainties in the normalization process. We have added an evaluation of this issue in the discussion section and included a note to caution readers about the potential uncertainties associated with this period of the dataset. The added description is as follows:

**4.1 Improvements in cross-sensor harmonization**
* * *
Secondly, our harmonization strategy uses GOME-2A as the reference sensor. Its extended data record (2007–2021) provides over five years of overlap with both SCIAMACHY and OCO-2, allowing us to perform a single-step normalization for each. This approach helps reduce the uncertainty propagation associated with multi-step corrections. In contrast, the LT_SIFc* product uses GOME as the benchmark, relying on only a six-month overlap with SCIAMACHY and then sequentially calibrating SCIAMACHY and GOME-2A, which may accumulate uncertainties.

To quantify the impact of overlap duration on harmonization uncertainty, we performed normalization experiments using 6-, 12-, and 24-month overlap periods between GOME-2 and OCO-2. Each experiment

https://doi.org/10.5194/essd-2025-94

was repeated 10 times to assess the variability of the resulting harmonized time series.

[Figure]

Figure 13. Normalized SIF time series using different overlap durations: 6 months (first column), 12 months (second column), and 24 months (third column) from SCIAMACHY (top row) and OCO-2 (bottom row). The shaded areas represent the standard deviation (SD) across multiple experiments, with units of mW m$^{-2}$ sr$^{-1}$ nm$^{-1}$.

The results show that a six-month overlap leads to a higher standard deviation in SIF time series compared to longer overlaps. As the overlap period was extended from 6 to 12 months, the standard deviations of the normalized SIF series decreased from 0.015 to 0.007 mW m$^{-2}$ sr$^{-1}$ nm$^{-1}$ (SCIAMACHY) and from 0.018 to 0.005 mW m$^{-2}$ sr$^{-1}$ nm$^{-1}$ (OCO-2), representing a reduction of over 53.3%. **These results confirm that short overlap periods increase normalization uncertainty and highlight the robustness of our chosen strategy, which avoids using GOME as the baseline. Nevertheless, users should be aware of the potential uncertainties in the early portion (from 1995 to 2003) of the LHSIF record.**

**Comment 3:** Line 21: You should better change a word to substitute 'tool' in line 21. How could a dataset be a tool? How about "Therefore, the long-term harmonized SIF dataset with a fine 0.05° resolution is valuable for estimating global photosynthesis over extended periods."

Response: Thanks for this comment. We have revised the sentence according to your advice.

**Comment 4:** Missing space in multiple places in sentences.

Response: Thanks for this comment. We have thoroughly reviewed the manuscript and made corrections.

**Comment 5**:Line 22: Missing space: "The LHSIF dataset is available at https://doi.org/10.5281/zenodo.14854185 (Zou et al., 2025)."
Response: Thanks for this comment. It has been corrected.

**Comment 6**:Line 138: There should be a comma in"0.5°×0.5°, and 1°×1°".
Response: Thanks for this comment. It has been corrected.

**Comment 7**:Line 144: GPP (Berry et al)
Response: Thanks for this comment. It has been corrected.

**Comment 8**:Line 153: Missing space : 2 m.
Response: Thanks for this comment. It has been corrected.

**Comment 9**:Line 154: Missing comma: f(NIRv), f(VPD), and f(AT),
Response: Thanks for this comment. It has been corrected.

**Comment 10**:Line 159: "L-BFGS-B" algorithm (Byrd et al., 1995)
Response: Thanks for this comment. We have added a space before the citation.

**Comment 11**:Line 167: Two balnk spaces here. "0.05 , were"
Response: Thanks for this comment. It has been corrected.

**Comment 12**:Line 445: Missing space before the reference.
Response: Thanks for this comment. It has been corrected.

**Comment 13**:Incomplete sentence in Line 267: "The temporal and spatial distributions of the spatial downscaling residuals were analyzed (Fig. 4). The residual was calculated as the difference between ? As shown..."
Response: Thanks for this comment. The missing explanation has been added as follows:

The residuals are calculated as the difference between the reaggregated SIF (SIF_reagg) and the original SIF (SIF_original).

**Comment 14**:L353: Please do grammer check and change this sentence to: "To highlight the overall correlation, the data in June for these two sites were removed from the scatter plots."
Thanks for this comment. We have revised the sentence according to your advice.

**Comment 15**:Line434: Upper foot label for m-2 in "residual less than 0.05 mW m-2 sr-1 nm-1".

Response: Thanks for this comment. It has been corrected.

**Comment 16**:Line 303: please add an 'and' in "0.05; and significant increase".

Response: Thanks for this comment. It has been corrected.

---

## Referee Report (RR1)

The authors have addressed nearly all of my major concerns. However, several minor issues remain, which I suggest the authors revise. While the use of pseudo-invariant calibration sites (e.g., deserts) is valuable for evaluating sensor radiometric stability, these regions contain little to no vegetation do not emit meaningful SIF signals. As such, analysis may not convincingly validate the reliability of interannual SIF trends. The dataset has clear value and the revision has improved the manuscript. However, I recommend a minor revision to address the following issue before acceptance.

Minor :

1)The authors only provide pseudo-invariant calibration site analyses to argue against sensor degradation. Yet, no independent validation is offered to demonstrate whether the strong decline during 1995-2000 reflects real ecosystem dynamics or methodological artifacts. Cross-validation with indepedent datasets (e.g., tree-ring records, AVHRR NDVI, FAPAR, or process-based model simulations) would be necessary to convince readers that these early trends are credible.

2)Some terms are over-stated (such as robust, significant (without P value), and substantial) without sufficient quantitative evidence. More cautious and objective wording is recommended. Here are some examples: Line 274: "Our dataset provides a robust estimate…"; Line 512: "…show a significant improvement compared with…"; Line 742: "…a substantial amount of carbon flux variability remains unexplained."

3)I recommend you to add all abbreviations for each figure. For example, in Fig. 10, please add the definations or full names for the following abbreviations (including LHSIF (red), LT_SIFc* (green), SIF_005 (purple), and LCSIF (blue)) at the end of the figure caption. In addition, using two stars for significance level is more common (i.e., * for $P<0.05$ and ** for $P<0.01$).

4)Please double check all figures to aviod typo errors.

Fig. 10(d): typo error: 'Coldl'——>"Cold";

Fig. 10(c), (e): typo error: "Temporate"——>"Temperate"

Fug. 10(e), (h): why they are the same name? Maybe the layout could be improved by putting of vegetations in different climate regions in the same row or line.

[Figure]

5)Line 22: "…has garnered significant attentions…" (change 'attentions' to 'attention').

6)Line 46-47: "…The Orbiting Carbon Observatory(OCO)-2 satellite…" (Missing space before parentheses).

7)Line 181–185: "…Eq. (1) can be broken down into three terms (Bacour et al., 2019): " (Replace the full-width colon with a standard English colon).

8)Line 309: Please ensure that the first word in x- and y-axis titles starts with a capital letter for consistency (e.g., Fig. 7a: "Yearly max SIF" instead of "yearly max SIF"). Same as in Fig. 8b.

---

## Author Response (AR2)

https://doi.org/10.5194/essd-2025-94
**Author's Response**

Dear editor and reviewers,

We sincerely appreciate your time and careful review of our work.Please find below the reviewers' comments (in black), followed by our responses (in blue), the contents of the manuscript (in purple), and the revised text (in red) in the manuscript.

In addition, in the revised version, we have adjusted the colour schemes of Figures 5 and 10 to improve accessibility for readers with colour vision deficiencies.

Please let us know if there is any additional information for your evaluation of the manuscript.

Best regards,
Chu Zou
zouchu20@mails.ucas.ac.cn

https://doi.org/10.5194/essd-2025-94
**Response to Reviewer 1 Comments:**

The authors have addressed nearly all of my major concerns. However, several minor issues remain, which I suggest the authors revise. While the use of pseudo-invariant calibration sites (e.g., deserts) is valuable for evaluating sensor radiometric stability, these regions contain little to no vegetation do not emit meaningful SIF signals. As such, analysis may not convincingly validate the reliability of interannual SIF trends. The dataset has clear value and the revision has improved the manuscript. However, I recommend a minor revision to address the following issue before acceptance.

We sincerely appreciate the reviewer's recognition of our work and the thoughtful suggestions provided. We have carefully addressed each of the comments. In response, we have conducted additional validation of the dataset and refined ambiguous terminology throughout the manuscript. Please find our detailed responses below.

**Comment 1:** The authors only provide pseudo-invariant calibration site analyses to argue against sensor degradation. Yet, no independent validation is offered to demonstrate whether the strong decline during 1995-2000 reflects real ecosystem dynamics or methodological artifacts. Cross-validation with indepedent datasets (e.g., tree-ring records, AVHRR NDVI, FAPAR, or process-based model simulations) would be necessary to convince readers that these early trends are credible.

**Response:** We appreciate the reviewer's helpful comment. The cross-sensor harmonization employed in this study can reduce inter-sensor biases but cannot fundamentally correct potential errors in the original GOME retrievals. The early GOME record (1995–2003) may be affected by sensor degradation, coarse spatial resolution (leading to mixed-pixel effects), low signal-to-noise ratio, and increased retrieval uncertainties in low-fluorescence or high-latitude regions (Burrows et al., 1999; Joiner et al., 2013; Köhler et al., 2015). In the revised manuscript, we added comparisons with other SIF products and AVHRR NDVI during this period. While some broad consistencies are observed, notable discrepancies remain, highlighting the need for cautious interpretation of early GOME-based trends. Future work will require dedicated strategies for validation and correction, such as radiometric recalibration using pseudo-invariant sites (Zou et al., 2024) or harmonization approaches based on physical mechanisms. These clarifications are reflected in the revised Section 4.2.

4.2  Limitations and future perspectives

\*\*\*\*\*\*

Although the normalization method was designed to minimize the influence of GOME-related uncertainties on the harmonized dataset, the accuracy of early LHSIF data (1995–2003) still warrants cautious interpretation. Additional analyses were conducted for the GOME observation period. Despite the brief overlap with SCIAMACHY, the two datasets showed broadly consistent seasonal dynamics (Figure 14a). We further compared the temporal trends of LHSIF, LCSIF, and AVHRR NDVI during 1995–2003 (Figure 14b–d). Some regions, such as western Europe, northern Oceania, and the southern parts of both North and South America, showed broadly consistent increasing trends across datasets. Conversely, declines were commonly observed in central Africa, southern Oceania, the Amazon rainforest, and northwestern India. Nevertheless, noticeable discrepancies remain. For instance, LHSIF displayed more extensive declines in high-latitude regions and central North America, which were not consistently captured by either LCSIF or NDVI.

https://doi.org/10.5194/essd-2025-94

[Figure]

Figure 14 Comparisons of GOME SIF, SCIAMACHY SIF, LHSIF, LCSIF, and AVHRR NDVI during 1995–2003. (a) Correlation coefficient between GOME and SCIAMACHY SIF time series; annual trends of (b) LHSIF, (c) LCSIF, and (d) AVHRR NDVI. The scatter points represent statistical significance($p < 0.05$).

These inconsistencies may reflect several limitations of the early GOME record, including (i) the coarse spatial resolution that amplifies mixed-pixel effects (Joiner et al., 2013), (ii) the relatively low signal-to-noise ratio of the GOME instrument (Burrows et al., 1999), (iii) increased retrieval uncertainties in high-latitude regions with low fluorescence intensity (Köhler et al., 2015), and (iv) potential uncorrected sensor degradation effects. As our harmonization approach primarily reduces inter-sensor biases through normalization, it cannot fundamentally resolve these intrinsic limitations of the original GOME data. In addition, errors may also arise from the propagation and accumulation of uncertainties during the normalization process, since GOME was further adjusted based on the corrected SCIAMACHY product.

Future work will require dedicated strategies to address the intrinsic limitations of early GOME observations. Such strategies may include radiometric recalibration using pseudo-invariant sites (Zou et al., 2024) and also physically-based harmonization approaches to mitigate sensor inconsistencies arising from observation geometry, atmospheric conditions, pixel size, and background signals. Implementing these approaches will enhance the reliability of early trends, providing a more robust foundation for interpreting long-term variations in satellite-observed SIF.

**Comment 2:** Some terms are over-stated (such as robust, significant (without P value), and substantial) without sufficient quantitative evidence. More cautious and objective wording is recommended. Here are some examples: Line 274: "Our dataset provides a robust estimate…"; Line 512: "…show a significant improvement compared with…"; Line 742: "…a substantial amount of carbon flux variability remains unexplained."

**Response:** Thanks for this comment. We noticed that the line numbers provided in the examples may not correspond exactly to the current version of the manuscript. Nevertheless, the entire manuscript was reviewed, and the potentially overstated terms were replaced with more cautious and objective wording. The revisions are as follows:

https://doi.org/10.5194/essd-2025-94

**Line 77:**

**Original Sentence:** This correction effectively eliminates the influence of sensor degradation over time, providing a robust benchmark for generating long-term harmonized SIF products.

**Revision:** This correction effectively eliminates the influence of sensor degradation over time, providing a practical reference for generating long-term harmonized SIF products.

**Line 302:**

**Original Sentence:** The MSD was significantly reduced after temporal correction, with a notable decrease in the proportion of bias.

**Revision:** The MSD was reduced after temporal correction, with a decrease in the proportion of bias.

**Line 309:**

**Original Sentence:** Significant interannual fluctuations were found for the SIF time series without normalization, with an overall decline (blue line in Fig. 7a).

**Revision:** Noticeable interannual fluctuations were found for the SIF time series without normalization, with an overall decline (blue line in Fig. 7a).

**Line 444:**

**Original Sentence:** The region-based approach allows for a larger sample size within each region, potentially enabling a more robust estimation of the CDF, while the month-specific treatment helps account for seasonal variations in the CDF.

**Revision:** The region-based approach allows for a larger sample size within each region, potentially enabling an improved estimation of the CDF, while the month-specific treatment helps account for seasonal variations in the CDF.

**Line 498:**

**Original Sentence:** Our analysis demonstrated that the harmonized dataset significantly reduced overall errors by more than 49% and exhibited a stable interannual increase ($0.31 \pm 0.07\%$ yr$^{-1}$).

**Revision:** Our analysis demonstrated that the harmonized dataset reduced overall errors by more than 49% and exhibited a stable interannual increase ($0.31 \pm 0.07\%$ yr$^{-1}$).

**Comment 3:** I recommend you to add all abbreviations for each figure. For example, in Fig. 10, please add the definations or full names for the following abbreviations (including LHSIF (red), LT_SIFc* (green), SIF_005 (purple), and LCSIF (blue)) at the end of the figure caption. In addition, using two stars for significance level is more common (i.e., * for P<0.05 and ** for P<0.01).

**Response:** We appreciate this suggestion. To ensure clarity while avoiding redundancy, we have consolidated all dataset definitions into a new summary table (Table 2), which includes descriptions, temporal coverage, sensor sources, and processing methods for each SIF product. This table is referenced in all relevant figure captions (e.g., 'See Table 2 for dataset details'), allowing readers to access complete information without repetitive text in captions.

**2.5 Datasets for validation and comparison analysis**

Multiple long-term satellite-derived products were utilized for cross-validation in this study. Key characteristics of these benchmark datasets, along with the proposed LHSIF product, are summarized in Table 2.

https://doi.org/10.5194/essd-2025-94

Table 2. The long-term products used in this study and the relevant details about them.

| Dataset | Description | Time coverage | Sensors for SIF product | Processing method |
|---|---|---|---|---|
| LHSIF | Multi-sensor harmonized SIF with extended temporal coverage | 1995.07–2024.12 | GOME, SCIAMACHY, GOME-2, OCO-2 | CDF matching |
| LT_SIFc* | Multi-sensor harmonized SIF | 1995.07–2018.12 | GOME, SCIAMACHY, GOME-2 | CDF matching |
| SIF_005 | Harmonized SIF | 2003.01–2017.12 | SCIAMACHY, GOME-2 | CDF matching |
| LCSIF | Spatially continuous reconstructed SIF | 1982.01–2023.12 | OCO-2 | Neural network |
| BEPS GPP | Simulated GPP using ecological process model | 1981.01–2019.12 | - | - |
| AVHRR NDVI | Long term NDVI product addressed for temporal inconsistency | 1982.01–2021.12 | - | - |
| AVHRR NIRv | Long term NIRv product addressed for temporal inconsistency | 1982.01–2021.12 | - | - |

Regarding the significance levels, the correlated figures (Figure 5 and Figure 10) were revised as follows:

[Figure]

Figure 5. Global-averaged SIF time series derived from GOME (yellow), SCIAMACHY (blue), GOME-2 (green), and OCO-2 (red), along with the long-term harmonized SIF time series (LHSIF, gray dotted line), which aligns the satellite datasets based on overlapping periods. The table on the right summarizes the statistical characteristics of each sensor, including the mean, standard deviation (std), and the annual trend (Trend) averaged over the respective periods. The statistical significance of the trends is indicated as follows: n.s. for not significant ($p \geq 0.05$), * for significant ($p < 0.05$), and ** for highly significant ($p < 0.01$).

https://doi.org/10.5194/essd-2025-94

[Figure]

Figure 10. Comparison of interannual variations in long-term SIF products. LHSIF (red), LT_SIFc* (green), SIF_005 (purple), and LCSIF (blue) are compared for (a) the global scale and (b–m) various climatic and vegetation regions. All datasets were normalized using the z-score method. Dashed lines represent yearly maximum values, and solid lines indicate linear trends. To aid visual comparison, trend lines were anchored at the origin (2010, 0). The statistical significance of the trends is indicated as follows: n.s. for not significant ($p \geq 0.05$), * for significant ($p < 0.05$), and ** for highly significant ($p < 0.01$). See Table 2 for dataset details.

**Comment 4:** Please double check all figures to aviod typo errors.

(1) Fig. 10(d): typo error: 'Coldl'——>"Cold";

**Response:** Thanks for this comment. It has been corrected.

(2) Fig. 10(c), (e): typo error: "Temporate"——>"Temperate"

**Response:** Thanks for this comment. It has been corrected.

(3) Fig. 10(e), (h): why they are the same name? Maybe the layout could be improved by putting of vegetation in different climate regions in the same row or line.

**Response:** Thanks for this comment. The figure has been reorganized, and the labeling inconsistencies have been corrected. The revised figure is as follows:

[Figure]

Figure 10. Comparison of interannual variations in long-term SIF products. LHSIF (red), LT_SIFc* (green), SIF_005 (purple), and LCSIF (blue) are compared for (a) the global scale and (b–m) various climatic and vegetation regions. All datasets were normalized using the z-score method. Dashed lines represent yearly maximum values, and solid lines indicate linear trends. To aid visual comparison, trend lines were anchored at the origin (2010, 0). The statistical significance of the trends is indicated as follows: n.s. for not significant ($p \geq 0.05$), * for significant ($p < 0.05$), and ** for highly significant ($p < 0.01$). See Table 2 for dataset details.

**Comment 5:** Line 22: "…has garnered significant attentions…" (change 'attentions' to 'attention').
**Response:** Thanks for this comment. It has been corrected.

**Comment 6:** Line 46-47: "…The Orbiting Carbon Observatory(OCO)-2 satellite…" (Missing space before parentheses).
**Response:** Thanks for this comment. It has been corrected.

**Comment 7:** Line 181–185: "…Eq. (1) can be broken down into three terms (Bacour et al., 2019):  " (Replace the full-width colon with a standard English colon).
**Response:** Thanks for this comment. It has been corrected.

https://doi.org/10.5194/essd-2025-94

**Comment 8:** Line 309: Please ensure that the first word in x- and y-axis titles starts with a capital letter for consistency (e.g., Fig. 7a: "Yearly max SIF" instead of "yearly max SIF"). Same as in Fig. 8b.

**Response:** Thanks for this comment. The figures have been corrected as below:

[Figure]

Figure 7. (a) Trend and (b) box plot of the yearly maximum global-averaged SIF of the combined time series before (SIF$_u$) and after (SIF$_c$) normalization during 1995–2024.

[Figure]

Figure 8. (a) Map of trends in LHSIF for 1996–2024. (b) Percentage of areas in global vegetation covered by four different trend types (significant decrease: negative correlation and $p < 0.05$; decrease: negative correlation and $p \geq 0.05$; increase: positive correlation and $p \geq 0.05$; and significant increase: positive correlation and $p < 0.05$). The black dots in (a) represent statistically significant trends ($p < 0.05$). The statistics begin in 1996 due to incomplete data coverage in 1995, which only includes the second half of the year (July–December).

https://doi.org/10.5194/essd-2025-94
**Response to Reviewer 2 Comments:**

The authors have conducted extensive experimental analyses and made substantial revisions in response to the review comments, which have helped address the main issues.

We sincerely thank the reviewer for the detailed and constructive comments. We have incorporated new validation and discussion based on the reviewer's feedback and corrected a few typing errors. Please see the details below.

Several minor revisions remain as follows:

**Comment 1:** In response to the previous major comment #1, I appreciate that the novelty of this paper lies in the corrected TCSIF dataset rather than the correction method itself. Still, I was wondering about the following:
(1) Could the authors elaborate a bit more on the reason for changing the matching method? While the advantages of the new method were briefly mentioned, it might be helpful to understand why it wasn't adopted initially.
**Response:** Thanks for this comment. Regarding the choice of data matching methods, both the moment-matching and CDF-based approaches have their advantages and limitations. The moment-matching method is simple to implement and computationally efficient, and it is relatively robust when the sample size is small. However, it cannot fully account for higher-order distribution characteristics, such as skewness and kurtosis. In contrast, CDF matching can align the entire distribution but requires a sufficiently large sample size. herefore, the choice between these methods depends critically on sample size.
The rationale for the method change is as follows:
  (1)  Initially, the CDF method was not adopted because in the first version of the manuscript, matching was performed at the pixel level. At this scale, the available data for CDF estimation were limited, so we opted for moment-matching method based on mean and standard deviation to estimate the overall distribution.
  (2)  In the revised version, we instead applied the CDF matching procedure within different ecological strata. Ecological stratification ensures CDF matching operates within environmentally homogeneous units. Besides, this approach substantially increases the sample size and allows a more reliable estimation of the CDF functions.
     In summary, the change in method was made to achieve a more accurate and stable representation of the distribution.
(2) The use of a stratified CDF method based on Köppen climate classification and land cover products, along with accounting for phenological changes, is noted. Given that the LHSIF data cover a relatively long time span, have the authors also considered how long-term land cover changes over the years might affect the results?
**Response:** Thanks for this comment. To address the potential impact of long-term land cover changes, we accounted for the variability of land cover by using time-dependent land cover products. In the revised manuscript, we have clarified the method as below:
2.3  CDF matching method
     The cross-sensor SIF normalization was implemented using a stratified CDF matching approach to account for environmental variability. Specifically, the stratification was done based on a combination of Köppen climate zones (Beck et al., 2023) and the MODIS land cover types product (MCD12C1; Friedl and

https://doi.org/10.5194/essd-2025-94

Sulla-Menashe, 2022). The land cover map of the central year within the overlapping period was used to construct the CDFs, while the CDFs were applied each year according to the yearly land cover types. For the period before 2001, when MCD12C1 data were unavailable, the land cover map of 2001 was applied. The degradation-corrected GOME-2A dataset was used as the normalization reference for all other satellite-derived SIF datasets, based on their overlapping periods. The normalization of GOME data was based on the SCIAMACHY dataset, which had been previously normalized with GOME-2 data.

This implementation ensured that long-term land cover changes were inherently incorporated into the normalization process. Small fluctuations in land cover or algorithmic uncertainties exert only a minor effect on the fitted distributions, and their impact on the long-term consistency of the harmonized SIF record is expected to be negligible. The limitation is also discussed in the revised manuscript:

4.2 Limitations and future perspectives

Our investigation shows that the CDF normalization approach effectively reduces disparities across sensors, providing a unified reference framework with the longest time series to date. While the normalized dataset exhibits consistent seasonal and interannual patterns across sensors, several methodological considerations warrant discussion. First, as a statistical approach distinct from physical calibration methods (e.g., pseudo-invariant target radiometry), CDF matching may retain minor sensor-specific biases. Second, although we incorporate annual land cover updates using MCD12C1 product to account for vegetation dynamics, inherent classification uncertainties in the reference dataset persist. Nevertheless, the percentile-based CDF matching demonstrates inherent robustness against outliers (Wang et al., 2022), rendering land cover-induced biases negligible in practice.

**Comment 2:** Regarding previous major comment #3, the authors used overlapping periods between GOME-2, SCIAMACHY, and OCO-2 to illustrate the influence of short overlap periods. The manuscript also appropriately highlights potential uncertainties in the early-stage data. However, I have some additional concerns:

(1) Would it be possible to provide a quantitative estimate or discussion of the accuracy or uncertainty associated with the GOME-phase product? Unlike SCIAMACHY and OCO-2, the GOME data were corrected after SCIAMACHY, which may introduce additional error propagation. The relatively short (six-month) overlap between GOME and SCIAMACHY could further contribute to uncertainty.

**Response:** Thanks for this helpful comment. We have strengthened the evaluation of the consistency between GOME and SCIAMACHY, and additionally compared LHSIF, LCSIF, and NIRv products during the early period (1995–2003). The early GOME-phase record is subject to relatively high uncertainty, and the cross-sensor normalization cannot fundamentally resolve potential degradation or biases in the original GOME retrievals. These inconsistencies may arise not only from errors inherent to GOME itself, but also from the propagation and accumulation of uncertainties during the normalization process. We have clarified these limitations in the revised Section 4.2 and highlighted potential directions for future improvement.

4.2 Limitations and future perspectives
* * *
Although the normalization method was designed to minimize the influence of GOME-related uncertainties on the harmonized dataset, the accuracy of early LHSIF data (1995–2003) still warrants cautious interpretation. Additional analyses were conducted for the GOME observation period. Despite the brief overlap with SCIAMACHY, the two datasets showed broadly consistent seasonal dynamics (Figure 14a). We further compared the temporal trends of LHSIF, LCSIF, and AVHRR NDVI during 1995–2003 (Figure 14b–d). Some regions, such as western Europe, northern Oceania, and the southern parts of both North and South America, showed broadly consistent increasing trends across datasets. Conversely, declines were commonly

observed in central Africa, southern Oceania, the Amazon rainforest, and northwestern India. Nevertheless, noticeable discrepancies remain. For instance, LHSIF displayed more extensive declines in high-latitude regions and central North America, which were not consistently captured by either LCSIF or NDVI.

[Figure]

Figure 14 Comparisons of GOME SIF, SCIAMACHY SIF, LHSIF, LCSIF, and AVHRR NDVI during 1995–2003. (a) Correlation coefficient between GOME and SCIAMACHY SIF time series; annual trends of (b) LHSIF, (c) LCSIF, and (d) AVHRR NDVI. The scatter points represent statistical significance($p < 0.05$).

These inconsistencies may reflect several limitations of the early GOME record, including (i) the coarse spatial resolution that amplifies mixed-pixel effects (Joiner et al., 2013), (ii) the relatively low signal-to-noise ratio of the GOME instrument (Burrows et al., 1999), (iii) increased retrieval uncertainties in high-latitude regions with low fluorescence intensity (Köhler et al., 2015), and (iv) potential uncorrected sensor degradation effects. As our harmonization approach primarily reduces inter-sensor biases through normalization, it cannot fundamentally resolve these intrinsic limitations of the original GOME data. In addition, errors may also arise from the propagation and accumulation of uncertainties during the normalization process, since GOME was further adjusted based on the corrected SCIAMACHY product.

Future work will require dedicated strategies to address the intrinsic limitations of early GOME observations. Such strategies may include radiometric recalibration using pseudo-invariant sites (Zou et al., 2024) and also physically-based harmonization approaches to mitigate sensor inconsistencies arising from observation geometry, atmospheric conditions, pixel size, and background signals. Implementing these approaches will enhance the reliability of early trends, providing a more robust foundation for interpreting long-term variations in satellite-observed SIF.

(2) In Figure 10, LHSIF and LCSIF show similar relative magnitudes and variations in the early period, in contrast to LTSIF_c*. This might serve as supporting evidence to strengthen confidence in early-stage SIF data, and could perhaps be discussed in the manuscript. BTW, in 10d (subpanel), there appears to be a typo: "coldl?". Please correct.

**Response:** We thank the reviewer for highlighting the potential comparison with early-stage data. The revisions are as below:

https://doi.org/10.5194/essd-2025-94

The results show that a six-month overlap leads to a higher standard deviation in SIF time series compared to longer overlaps. As the overlap period was extended from 6 to 12 months, the standard deviations of the normalized SIF series decreased from 0.015 to 0.007 mW m$^{-2}$ sr$^{-1}$ nm$^{-1}$ (SCIAMACHY) and from 0.018 to 0.005 mW m$^{-2}$ sr$^{-1}$ nm$^{-1}$ (OCO-2), representing a reduction of over 53.3%. These results confirm that short overlap periods increase normalization uncertainty and highlight the robustness of our chosen strategy, which avoids using GOME as the baseline. Besides, the early-stage LHSIF exhibits consistency with LCSIF (Fig. 10a), providing additional support for its early-period reliability.

The figure captions have been standardized, the panels have been reordered according to climate zones and land-cover types, and the original typographical errors have been corrected. The revised figure is provided below.

[Figure]

Figure 10. Comparison of interannual variations in long-term SIF products. LHSIF (red), LT_SIFc* (green), SIF_005 (purple), and LCSIF (blue) are compared for (a) the global scale and (b–m) various climatic and vegetation regions. All datasets were normalized using the z-score method. Dashed lines represent yearly maximum values, and solid lines indicate linear trends. To aid visual comparison, trend lines were anchored at the origin (2010, 0). The statistical significance of the trends is indicated as follows: n.s. for not significant (p ≥ 0.05), * for significant (p < 0.05), and ** for highly significant (p < 0.01).

**Comment 3:** Concerning previous major comment #5, the authors mention that LHSIF can more directly reflect regional responses to environmental variability compared to LCSIF. It would be helpful to include more

supporting evidence on this point. For example, in Figure 10a, could the authors highlight specific years where LCSIF and LHSIF differ noticeably, and relate these differences to known climatic events or anomalies? This could help illustrate the advantages of LHSIF more clearly.

**Response:** We thank the reviewer for this valuable suggestion. Satellite-observed SIF has been shown to capture ecosystem responses to major climatic events, such as the suppression of photosynthesis during the 2015/16 El Niño in the tropics and drought-induced declines in Europe and North America (Shekhar et al., 2020; Sun et al., 2015; Yoshida et al., 2015). Our study builds directly on these observational SIF datasets, thereby retaining their advantages in reflecting environmental variability. Therefore, in principle, LHSIF is expected to capture climate-driven events more effectively than LCSIF, which is based on data-driven method.

While differences between LHSIF and LCSIF may partially related to climate anomalies, this should not be interpreted as definitive evidence. Firstly, interannual fluctuations in SIF products arise from multiple sources, including retrieval uncertainties and residual atmospheric effects. Therefore, annual variations cannot be directly attributed to climatic events. Secondly, extreme climatic events may not be fully represented in the global mean values. Robust attribution would require regionally explicit analyses, which are beyond the scope of this study.

Although Figure 10 does not highlight climate events in specific years, the overall trends reflect the advantages of LHSIF. Consequently, instead of highlighting specific years in Figure 10a, we have expanded the discussion in Section 4.2 to emphasize the benefits of observation-based SIF products and to encourage future studies to examine the relationship between LHSIF and climatic events in a more regionally explicit manner.

**4.2  Limitations and future perspectives**
* * *
Another type of long-term SIF datasets have been generated by temporally extrapolating SIF observations based on machine-learning methods. These datasets provide more than two decades of high-temporal-resolution data beyond the monthly scale (Zhang et al., 2018b; Li and Xiao, 2019; Fang et al., 2023). However, such datasets predominantly depend on model-driven predictions constrained by satellite observation periods, rather than being based on actual observational data, which is fundamentally different from the approach employed here (Chen et al., 2025; Ma et al., 2022).

Previous findings have demonstrated that satellite-observed SIF is capable of capturing ecosystem responses to major climatic extremes, such as the suppression of photosynthesis during the 2015/16 El Niño event in the tropics and drought-induced declines in Europe and North America (Shekhar et al., 2020; Sun et al., 2015; Yoshida et al., 2015). These findings provide support for the potential advantages of LHSIF in reflecting regional environmental variability. Nevertheless, detailed attribution of interannual differences among products to specific climatic events requires dedicated analyses and applications, which should be pursued in future work. Currently, the temporal resolution of purely observation-based enhanced SIF products that span longer than 20 years remains constrained at the monthly scale, largely due to noise in the satellite SIF products. Overcoming this limitation will require further refinement of existing downscaling models, paving the way for future products to achieve a resolution of 16 days or higher.

**Comment 4:** In Detailed Comments #14 and #15, the publication year for the reference "Du" is inconsistent. Please verify and correct for consistency.

Response: Thanks for this comment. All citations have now been verified and standardized to "Du et al., 2023".

The references cited in the response are listed below:

https://doi.org/10.5194/essd-2025-94

Burrows, J. P., Weber, M., Buchwitz, M., Rozanov, V., Ladstätter-Weißenmayer, A., Richter, A., DeBeek, R., Hoogen, R., Bramstedt, K., Eichmann, K.-U., Eisinger, M., and Perner, D.: The Global Ozone Monitoring Experiment (GOME): Mission Concept and First Scientific Results, Journal of the atmospheric sciences, 56, 151–175, https://doi.org/10.1175/1520-0469, 1999.

Chen, S., Liu, L., Sui, L., Liu, X., and Ma, Y.: An improved spatially downscaled solar-induced chlorophyll fluorescence dataset from the TROPOMI product, Scientific Data, 12, 135, https://doi.org/10.1038/s41597-024-04325-6, 2025.

Ma, Y., Liu, L., Liu, X., and Chen, J.: An improved downscaled sun-induced chlorophyll fluorescence (DSIF) product of GOME-2 dataset, European Journal of Remote Sensing, 55, 168–180, https://doi.org/10.1080/22797254.2022.2028579, 2022.

Shekhar, A., Chen, J., Bhattacharjee, S., Buras, A., Castro, A. O., Zang, C. S., and Rammig, A.: Capturing the Impact of the 2018 European Drought and Heat across Different Vegetation Types Using OCO-2 Solar-Induced Fluorescence, Remote Sensing, 12, https://doi.org/10.3390/rs12193249, 2020.

Sun, Y., Fu, R., Dickinson, R., Joiner, J., Frankenberg, C., Gu, L., Xia, Y., and Fernando, N.: Drought onset mechanisms revealed by satellite solar-induced chlorophyll fluorescence: Insights from two contrasting extreme events, Journal of Geophysical Research: Biogeosciences, 120, 2427–2440, https://doi.org/10.1002/2015JG003150, 2015.

Yoshida, Y., Joiner, J., Tucker, C., Berry, J., Lee, J.-E., Walker, G., Reichle, R., Koster, R., Lyapustin, A., and Wang, Y.: The 2010 Russian drought impact on satellite measurements of solar-induced chlorophyll fluorescence: Insights from modeling and comparisons with parameters derived from satellite reflectances, Remote Sensing of Environment, 166, 163–177, https://doi.org/10.1016/j.rse.2015.06.008, 2015.